# The transcription factor OsSUF4 interacts with SDG725 in promoting H3K36me3 establishment

Bing Liu[1], Yuhao Liu[1], Baihui Wang[1], Qiang Luo[1], Jinlei Shi[1], Jianhua Gan[2], Wen-Hui Shen [1,3], Yu Yu[1] & Aiwu Dong[1]

The different genome-wide distributions of tri-methylation at H3K36 (H3K36me3) in various species suggest diverse mechanisms for H3K36me3 establishment during evolution. Here, we show that the transcription factor OsSUF4 recognizes a specific 7-bp DNA element, broadly distributes throughout the rice genome, and recruits the H3K36 methyltransferase SDG725 to target a set of genes including the key florigen genes *RFT1* and *Hd3a* to promote flowering in rice. Biochemical and structural analyses indicate that several positive residues within the zinc finger domain are vital for OsSUF4 function in planta. Our results reveal a regulatory mechanism contributing to H3K36me3 distribution in plants.

[1] Department of Biochemistry, State Key Laboratory of Genetic Engineering, Collaborative Innovation Center for Genetics and Development, International Associated Laboratory of CNRS-Fudan-HUNAU on Plant Epigenome Research, Institute of Plant Biology, School of Life Sciences, Fudan University, 200438 Shanghai, China. [2] Department of Physiology and Biophysics, School of Life Sciences, Fudan University, 200438 Shanghai, China. [3] Universite de Strasbourg, CNRS, IBMP UPR 2357, F-67000 Strasbourg, France. Correspondence and requests for materials should be addressed to Y.Y. (email: yuy@fudan.edu.cn) or to A.D. (email: aiwudong@fudan.edu.cn)

Histone lysine methylation, as one of the most studied epigenetic marks, is evolutionarily conserved and plays an essential role in regulating gene expression in eukaryotes, ranging from single-celled yeast to multicellular animals and plants. The presence of methylation on different lysine residues of histones and the extent (mono, di-, and tri-) of methylation represent distinct chromatin statuses that may subsequently affect the accessibility of protein factors to target DNA to promote or repress gene transcription[1]. Although histone lysine methylations are conserved epigenetic marks, their genomic distributions vary broadly across species, especially between animals and plants. For example, in mammals, tri-methylation at H3K9 (H3K9me3) is found in silenced chromatin and pericentromeric hetero-chromatin[2], while very low levels of H3K9me3 have been detected in plants; instead, H3K9me2 is mainly distributed in pericen-tromeric heterochromatin and at repeats and transposons within the euchromatin[3]. Although H3K4me3 is similarly enriched at the promoter regions of both animals and plants[4–6], H3K36me3 distributions are diverse. In animal cells, H3K36me3 is mainly distributed at the 3′ end of the gene body[7], but it is close to the transcription start site (TSS) in plants[8,9]. It suggests that different mechanisms for H3K36me3 establishment and function may exist between plants and animals.

Histone lysine methylation marks are mostly established by the SET-domain group (SDG) of proteins, and different classes of SDGs are responsible for depositing methyl on specific his-tone lysine residues[10–14]. For example, some of the Ash1 class SDGs are H3K36-specific histone methyltransferases (HMTa-ses), among which typical members include the yeast SET2 and human SETD2/HYPB[13]. At least three Ash1 class members both in Arabidopsis (SDG4/ASHR3, SDG8/ASHH2/EFS, and SDG26/ASHH1) and in rice (SDG708, SDG724, and SDG725) have been identified to be responsible for H3K36 methylation[8]. SDG4 is specifically expressed in floral organs and contributes to the regulation of pollen tube growth[15]. SDG26, SDG708, and SDG724 are involved in flowering time control[8,16–18]. SDG8 and SDG725 seem to be the major H3K36 HMTases in Arabidopsis and rice, respectively, because they are broadly expressed and involved in diverse biological processes[9,18–26]. Arabidopsis SDG8 and rice SDG725 are the closest homologs of yeast SET2 and human SETD2, and they share similar domain organization. This evolutionary conservation in modifying enzymes seems to conflict with the divergence in genomic distributions of H3K36me3, which prompted us to study the mechanisms underlying the establishment of H3K36 methyla-tion in various species.

Here, we use the rice H3K36 methyltransferase SDG725 as a bait to screen its binding proteins and identify a $C_2H_2$-type zinc finger transcription factor, named SUPPRESSOR OF FRI 4 (OsSUF4), as a binding protein of SDG725. OsSUF4 physically interacts with SDG725 and recognizes a 7-bp DNA element within the promoter regions of the rice florigen genes RICE FLOWERING LOCUS T1 (RFT1) and Heading date 3a (Hd3a) to promote rice flowering. The target genes of OsSUF4 are not limited to RFT1 and Hd3a but include a number of genes involved in many biological processes. Thus, our findings highlight that distinct binding proteins of histone-modifying enzymes may lead to divergence in histone methylation distribution and eventually exhibit individual functions in various species.

## Results

**Genome-wide distributions of H3K36me3 in various species.** We previously reported differences in the H3K36me3 distribution pattern in rice from that in animals[26]. To gain a complete understanding of H3K36me3 distributions in eukaryotes, we used published ChIP-sequencing (ChIP-seq) data of single-celled yeast, the invertebrates Caenorhabditis elegans and Drosophila mela-nogaster, the vertebrates mice and humans, and the model plants Arabidopsis thaliana and rice. The profiles and heatmaps of H3K36me3 across the different genomes are presented in Fig. 1. In yeast and C. elegans, H3K36me3 showed a similar pattern and was distributed over the entire gene body regions. In contrast, the distribution of H3K36 in D. melanogaster, mice, and humans was mainly enriched at the 3′ end of the gene body, with an apparent peak close to the transcription terminal site (TTS). In Arabidopsis and rice, H3K36me3 signals tended to accumulate at the 5′ end of the gene body, peaking near the TSS (Fig. 1). The different dis-tribution patterns of H3K36me3 in various species indicate the complicated and divergent mechanisms for establishing H3K36 methylation during eukaryotic evolution.

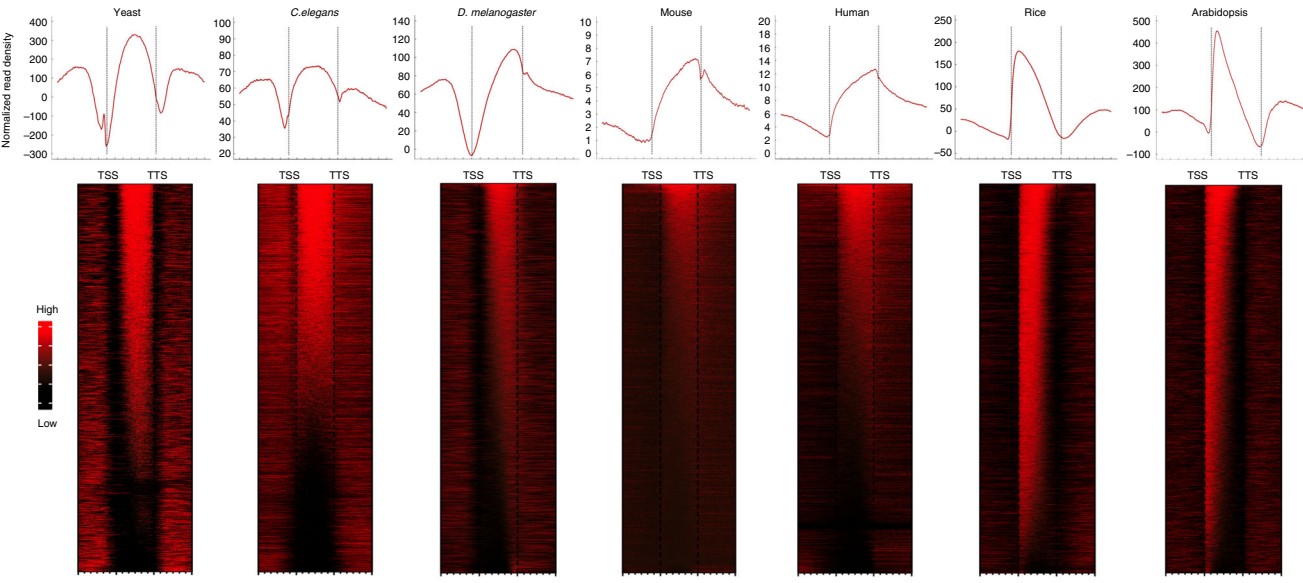

**Fig. 1** Genome-wide distributions of H3K36me3 in various species. Upper panel: integrative genomic distribution of H3K36me3 in yeast, C. elegans, D. melanogaster, mice, humans, rice, and Arabidopsis. Lower panel: corresponding heatmaps of H3K36me3. Each line represents a gene

**SDG725 interacts with transcription factor OsSUF4.** To determine why H3K36me3 was mainly distributed at the 5′ end of the gene body in plants, we chose SDG725, the major H3K36 methyltransferase in rice, as a bait to screen its binding proteins using the yeast two-hybrid system. Given that the full-length SDG725 protein was poorly expressed in yeast, the C-terminal fragment of SDG725 (1240–2150 amino acids, hereafter called SDG725C) containing the SET domain was chosen as the bait. Approximately 259 positive clones were obtained from yeast two-hybrid screening, of which 10 corresponded to *OsSUF4* (*Os09g38790*). *OsSUF4* encodes a $C_2H_2$-type zinc finger transcription factor[27,28]. Based on the amino acid sequence of full-length OsSUF4, we performed a phylogenetic analysis in eukaryotes and found that its homologs were widespread in eukaryotes but absent in yeast (Supplementary Fig. 1). The plant SUF4 proteins separated from other eukaryotes and

formed a group containing two distinct clades for monocot and dicot.

We performed yeast two-hybrid (Fig. 2a), glutathione S-transferase (GST) pulldown (Fig. 2b), and bimolecular fluorescence complementation (BiFC; Fig. 2c) assays to confirm the interaction between OsSUF4 and SDG725. In yeast, only cells expressing both OsSUF4 and SDG725 activated the *ADE2* reporter gene (Fig. 2c). For GST pulldown, we generated transgenic rice plants over-expressing HA-tagged OsSUF4 under the control of the maize ubiquitin promoter ($P_{Ubi}$::HA-SUF4) with a wild-type phenotype. GST pulldown experiments showed that HA-OsSUF4 protein was retained by GST-SDG725C but not by GST (Fig. 2b, Supplementary Fig. 2). For BiFC, the coding regions of *OsSUF4* and *SDG725C* were fused with the N-terminal and C-terminal fragments of *YFP*, respectively, and the resulting constructs *OsSUF4-YN* and *SDG725C-YC* were co-transfected

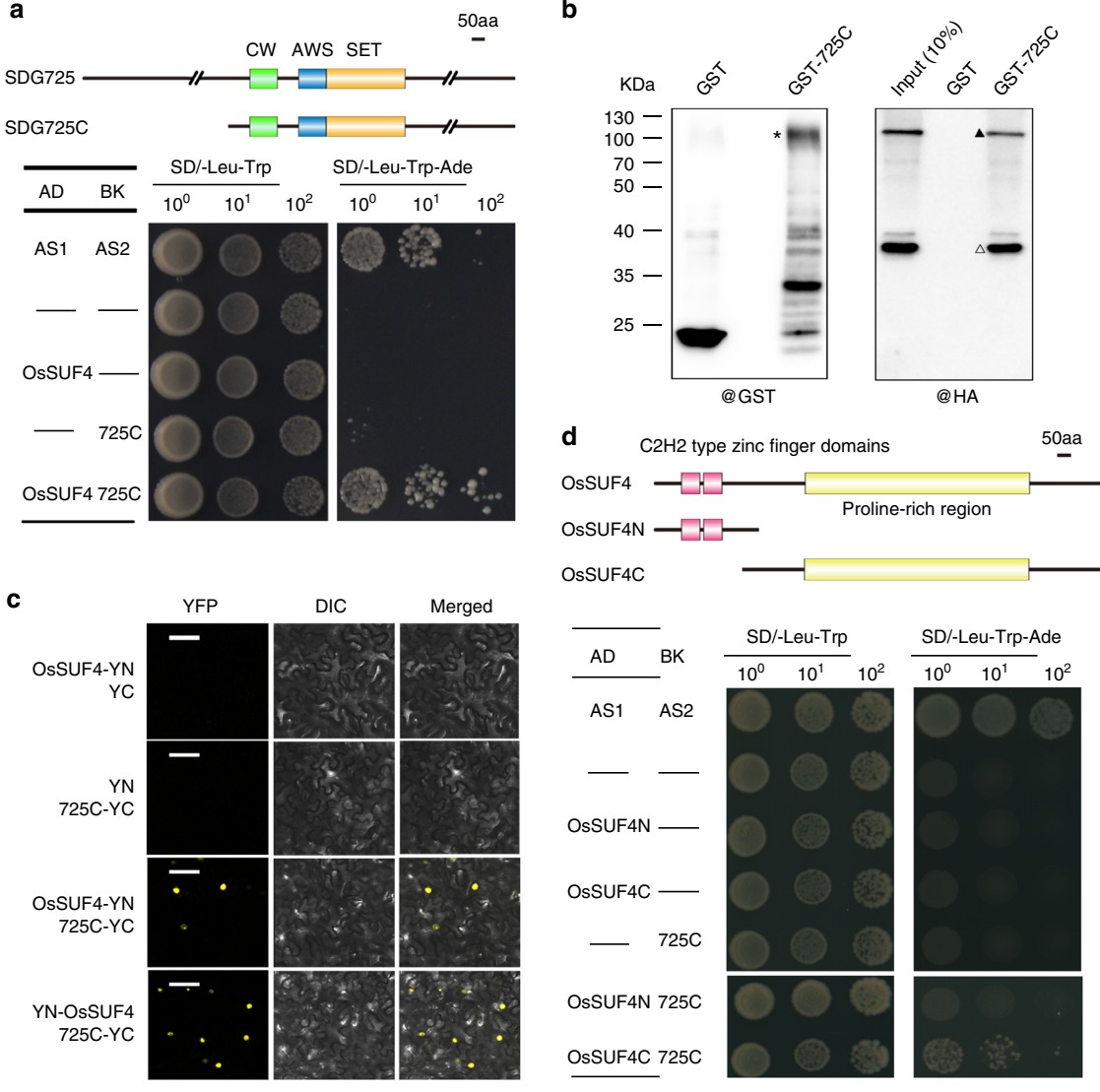

**Fig. 2** OsSUF4 interacts with SDG725C in vitro and in vivo. **a** The interaction between OsSUF4 and SDG725C in a yeast two-hybrid assay. AS1 and AS2 serve as the positive control. Upper panel: Schematic representation of full-length and truncated SDG725 proteins. **b** The left panel showed GST and GST-SDG725C (GST-725C) proteins used in the pulldown assay, and the right panel showed HA-OsSUF4 pulled down by SDG725C using rice plants over-expressing *HA-OsSUF4*. GST-725C, HA-OsSUF4, and a non-specific band are marked with an asterisk, a triangle, and an open triangle, respectively. **c** BiFC experiments showing the interaction of OsSUF4 and SDG725C in *Nicotiana benthamiana* leaf epidermal cells (Bar = 50 μm). DIC differential interference contrast, YN N-terminal of YFP, YC C-terminal of YFP. **d** Yeast two-hybrid findings that the C-terminal domain of OsSUF4 (OsSUF4C) interacts with SDG725C. Upper panel: Schematic representation of full-length and truncated OsSUF4 proteins. AS1 and AS2 serve as the positive control. Source data are provided as a Source Data file

into tobacco leaves. YFP signals were only observed in the nuclei when *OsSUF4-YN* and *SDG725C-YC* were co-transfected, indicating that OsSUF4 interacts with SDG725C in planta (Fig. 2c).

OsSUF4 comprises two conserved $C_2H_2$-type zinc finger domains at the N-terminus and a large proline-rich domain at the C-terminus. To investigate which domain of OsSUF4 is required for the interaction with SDG725C, we generated two constructs: one containing two zinc finger domains (1–105 amino acids, SUF4N), and the other including the proline-rich domain (58–355 amino acids, SUF4C). We found that SUF4C not SUF4N interacted with SDG725C in yeast two-hybrid experiments (Fig. 2d). Together, these results proved that the transcription factor OsSUF4 physically interacts with the H3K36-specific methyltransferase SDG725 in rice.

**OsSUF4 promotes rice flowering**. To investigate the function of OsSUF4 in vivo, we first checked its expression pattern in rice. Quantitative reverse transcription (qRT)-PCR analysis indicated that *OsSUF4* is transcribed ubiquitously, including in the root, stem, shoot, flag leaf, young leaf, and inflorescence of wild-type rice (Supplementary Fig. 3). RNA interference (RNAi) was subsequently performed to obtain a knockdown mutant of *OsSUF4*. Nucleotides 23–359 and 363–612 were, respectively, selected as

hairpin structures, resulting in two RNAi constructs $P_{35S}$::*OsSUF4Ri-1* (hereafter named *suf4Ri-1*) and $P_{35S}$::*OsSUF4Ri-2* (*suf4Ri-2*). A total of 20 and 15 independent transgenic lines of *suf4Ri-1* and *suf4Ri-2* were, respectively, obtained, which showed similar and stable phenotypes after five generations (Fig. 3a). qRT-PCR analysis confirmed that the transcription levels of *OsSUF4* in *suf4Ri-1* and *suf4Ri-2* dramatically decreased compared with wild-type plants (Fig. 3b), and *OsSUF4*-knockdown mutants displayed a late-flowering phenotype at both long day (LD) and short day (SD) conditions (Fig. 3c). As well as the RNAi lines, more than 10 stable co-suppression lines (hereafter referred to as *SUF4cs*) were obtained by transforming $P_{35S}$::*OsSUF4* into wild-type rice, which also showed decreased *OsSUF4* transcription and a late-flowering phenotype similar to *suf4Ri-1* and *suf4Ri-2* (Fig. 3a–c).

Coordinately, the late-flowering phenotype was observed in the *OsSUF4*-knockdown mutant *suf4Ri-1*, the *SDG725*-knockdown mutant *725Ri-1*, and the *suf4Ri-1 725Ri-1* double mutant (Fig. 3d–f), suggesting that both OsSUF4 and SDG725 function in the same genetic pathway of flowering time control in rice. Many rice genes have been identified as being involved in flowering time control, such as the two florigen genes *RFT1* and *Hd3a*, and their regulatory genes *Grain number, plant height, and*

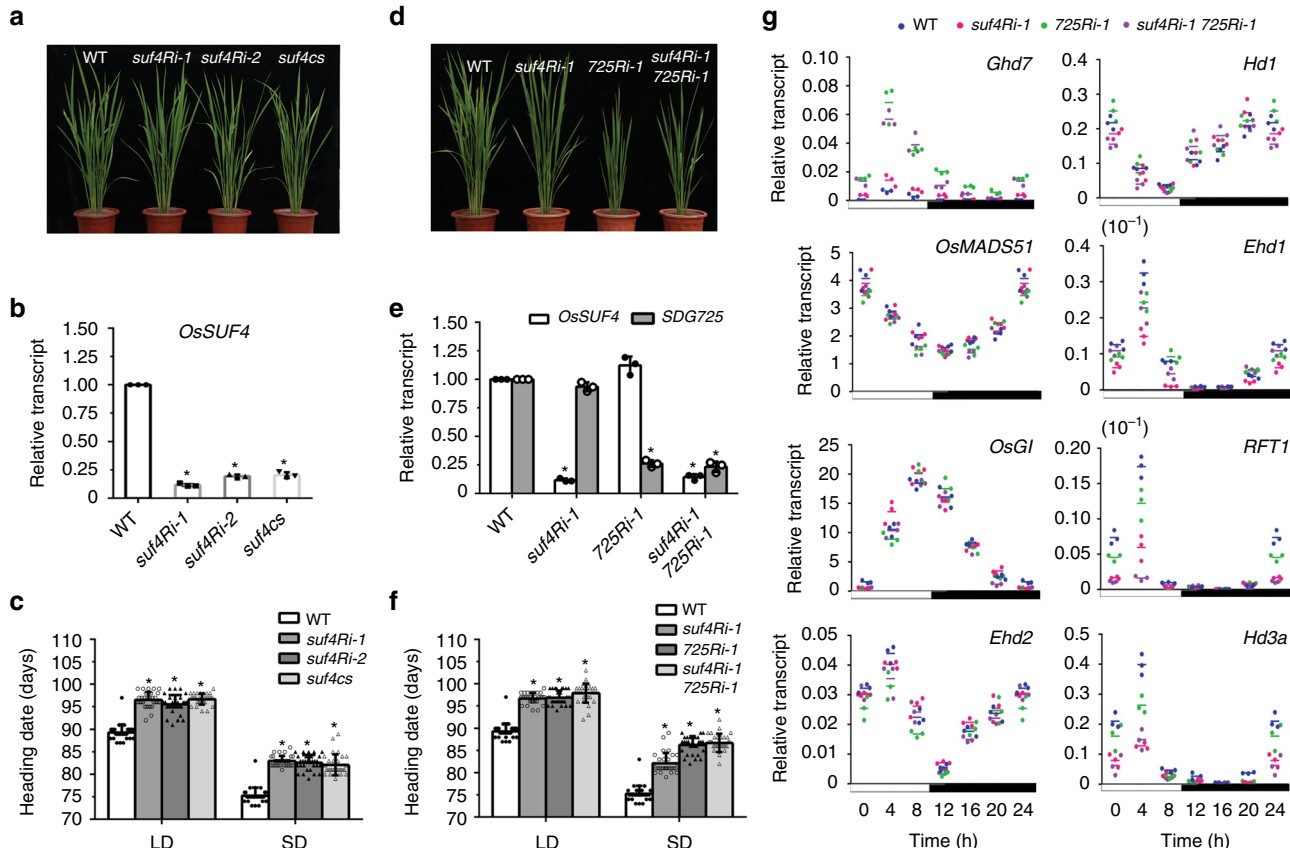

**Fig. 3** OsSUF4 promotes rice flowering under both LD and SD conditions. **a** Overall morphologies of 10-week-old wild-type (WT), two independent *OsSUF4* RNAi lines *suf4Ri-1* and *suf4Ri-2*, and *OsSUF4* co-suppressed (*suf4cs*) rice plants under SD conditions. **b** Relative transcript levels of *OsSUF4* in WT, *suf4Ri-1*, *suf4Ri-2*, and *suf4cs* plants under SD conditions. **c** Heading date of WT, *suf4Ri-1*, *suf4Ri-2*, and *suf4cs* plants grown in paddy fields under LD (Shanghai) and SD (Sanya) conditions. **d** Phenotypes of 10-week-old WT plants, and *suf4Ri-1*, *725Ri-1*, and *suf4Ri-1 725Ri-1* double mutants under SD conditions. **e** Relative transcript levels of *OsSUF4* and *SDG725* in WT, *suf4Ri-1*, *725Ri-1*, and *suf4Ri-1 725Ri-1* plants under SD conditions. **f** Heading date of WT, *suf4Ri-1*, *725Ri-1*, and *suf4Ri-1 725Ri-1* plants grown in paddy fields under LD (Shanghai) and SD (Sanya) conditions. **g** Diurnal transcription patterns of flowering genes in indicated plants under SD conditions. 30-day-old plants were sampled at intervals of 4 h from the start of illumination. For **c**, **e**, and **g** values are the mean ± standard deviation (SD) from three independent biological replicates normalized to the internal control OsUbiquitin5. Asterisks indicate significant differences between WT and mutants (Student's *t*-test: *$P < 0.01$). For **d** and **f** values shown are the mean ± SD of 30 individual plants. Asterisks indicate significant differences between WT and mutants (Student's *t*-test: *$P < 0.01$). Source data are provided as a Source Data file

*heading date 7* (*Ghd7*), *OsMADS51*, *GIGANTEA* (*OsGI*), *Ehd2*/*Rice INDETERMINATE 1* (*OsID1*/*RID1*), *Heading date 1* (*Hd1*), and *Early heading date 1* (*Ehd1*)[29].

We then traced the transcriptional change of these flowering genes in *OsSUF4*-knockdown mutants. Leaves of 30-day-old plants under SD conditions were collected at 4-h intervals over a 24-h period for RNA extraction and qRT-PCR. In *OsSUF4*-knockdown mutants *suf4Ri-1*, *suf4Ri-2*, and *suf4cs*, the expression of flowering genes was similar, so only that in *suf4Ri-1* is shown. The transcription levels of florigen genes *Hd3a* and *RFT1* and their upstream regulatory gene *Ehd1* were dramatically decreased upon knockdown of *OsSUF4* or *SDG725* (Fig. 3g). Similar to each single mutant, the transcription levels of these three flowering genes in the *suf4Ri-1 725Ri-1* double mutant were also significantly reduced (Fig. 3g). In contrast, the transcription levels of *OsMADS51*, *OsGI*, *Ehd2*, and *Hd1* were not obviously changed (*p*-value > 0.05) in *suf4Ri-1*, *725Ri-1*, and *suf4Ri-1 725Ri-1* (Fig. 3g). In agreement with the previous studies showing that *Ghd7* functions as a repressor of flowering[30], the transcripts of *Ghd7* were elevated in *suf4Ri-1*, *725Ri-1*, and *suf4Ri-1 725Ri-1* (Fig. 3g). Under LD conditions, *Hd3a*, *RFT1*, and *Ehd1* were also remarkably down-regulated in single mutants and the double mutant, and their transcription levels at 4 h after dawn are shown in Supplementary Fig. 4. Taken together, our results indicated that OsSUF4 promotes flowering in rice probably by affecting the expression of key flowering genes *Hd3a*, *RFT1*, and *Ehd1*.

**RFT1 and Hd3a are direct target genes of OsSUF4.** OsSUF4 is predicted to be a transcription factor, so we next questioned whether it directly binds to key flowering genes. ChIP-PCR was performed to analyze the possible in vivo binding of OsSUF4 to the flowering genes with a monoclonal antibody against OsSUF4. As shown in Supplementary Fig. 5a, the antibody recognized the purified his-tagged OsSUF4 but not the control His-FCA protein. The antibody specificity was further verified using rice plants overexpressing *HA-OsSUF4*. HA-OsSUF4 was successfully detected by both the HA antibody and OsSUF4 antibody (Supplementary Fig. 5b). ChIP-PCR analysis showed that OsSUF4 enrichment at fragments 2 (Nucleotides −1254 to −1155) and 3 (Nucleotides −503 to −426) of *RFT1* promoter and fragments 10 (Nucleotides −1038 to −937) and 11 (Nucleotides −562 to −477) of *Hd3a* promoter was significantly decreased in the *suf4Ri-1* mutant compared with the wild type (Fig. 4a). OsSUF4 enrichment was not changed at the loci within *Ehd1* and *MADS51* promoters, but *Ehd1* was down-regulated in *suf4Ri-1* (Supplementary Fig. 6), suggesting that *RFT1* and *Hd3a*, but not *Ehd1*, are the direct targets of OsSUF4. Notably, loss of SDG725 in the *725Ri-1* mutant reduced OsSUF4 enrichment at *RFT1* and *Hd3a* promoters; conversely, loss of OsSUF4 led to decreased SDG725 enrichment at *RFT1* and *Hd3a* promoters in the *suf4Ri-1* mutant (Fig. 4a), demonstrating that OsSUF4 and SDG725 enhance the binding of each other to *RFT1* and *Hd3a* promoters. When an antibody against H3K36me3 was used for ChIP-PCR analysis, most enrichment of H3K36me3 on *RFT1* and *Hd3a* chromatin was located downstream of the TSS (fragment 5 for *RFT1* and fragment 13 for *Hd3a*), which is consistent with the genome-wide distribution pattern of H3K36me3 in Fig. 1. However, H3K36me3 enrichment was also observed upstream of the TSS (fragment 4 for *RFT1* and fragment 12 for *Hd3a*). Consistent with the reduced enrichment of SDG725 at *RFT1* and *Hd3a* promoters, reduced H3K36me3 was also detected in the *suf4Ri-1* mutant (Fig. 4a). By using an antibody against H3, similar profiles of H3 enrichment on *RFT1*/*Hd3a* were observed in the wild type, the single mutants and the double mutant (Supplementary Fig. 7), indicating that the decreased levels of

OsSUF4, SDG725, and H3K36me3 at *RFT1*/*Hd3a* promoter regions in *suf4Ri-1* and *725Ri-1* mutants are not due to the loss of nucleosome occupancy.

To further investigate the genetic relationship between *OsSUF4* and *RFT1*/*Hd3a*, we generated the mutants in *RFT1* and *Hd3a* loci, respectively, by clustered regularly interspaced short palindromic repeats (CRISPR)/CRISPR-associated protein (Cas) 9[31]. The single guide RNAs (sgRNAs) were designed to target the specific sites within the 5′ ends coding regions of *RFT1* (5′-GAC CCAACAGCCCAGGGTCGTGG-3′) and *Hd3a* (5′-GGCTCCAA GACCGTGTCCAATGG-3′), respectively. After genotyping and sequencing, we confirmed to obtain four and five mutant lines for *RFT1* and *Hd3a*, respectively, which contained deletions or insertions within the coding region and resulted in early stop codons. Then we selected the line named *rft1-1* (containing 1-bp thymine insertion within the coding region) and the line *hd3a-1* (containing 1-bp adenine insertion within the coding region) for further analyses. Consistent with the late-flowering phenotype of known *RFT1*/*Hd3a* RNAi or CRISPR/Cas9 mutants[32,33], our *rft1-1* and *hd3a-1* mutants also flowered late under LD or SD conditions (Fig. 4b, c). We then generated *suf4Ri-1 rft1-1* and *suf4Ri-1 hd3a-1* double mutants by genetic crossing, which displayed late-flowering phenotypes similar to those of single mutants *rft1-1* and *hd3a-1* under LD or SD conditions, supporting the concept that *OsSUF4* acts upstream of *RFT1* and *Hd3a* to promote rice flowering (Fig. 4b, c).

**OsSUF4 binds to a specific 7-bp element.** Electrophoretic mobility shift assay (EMSA) was performed to refine the precise OsSUF4-binding element within the *Hd3a* promoter. First, we proved that OsSUF4 bound to fragment 10 of the *Hd3a* promoter (Nucleotides −1038 to −937, upstream of the TSS, hereafter named as *Hd3a-a*) in ChIP–PCR analysis (Fig. 4a). Thus, different truncated fragments of *Hd3a-a* were synthesized as DNA probes in EMSA experiments (Fig. 5a), showing that His-OsSUF4 bound to *Hd3a-c* but not to *Hd3a-b* (Fig. 5b). Next, we proved that His-OsSUF4N containing the $C_2H_2$ zinc finger domains, not His-OsSUF4C, bound to *Hd3a-c* (Fig. 5b). Then, *Hd3a-c* was shortened to *Hd3a-d* and *Hd3a-e*, and the 32-bp *Hd3a-e* fragment was finally determined as the effective OsSUF4-binding fragment (Fig. 5b).

To map the precise element, we generated a series of mutations within *Hd3a-e* and found that a 9-bp element (5′-TACGGAAA T-3′) is essential for the association with His-OsSUF4N (Fig. 5c). Consistently, *Hd3a*-mu8, in which the 9-bp element was mutated, failed to be recognized by His-OsSUF4N (Fig. 5d). We next searched for the 9-bp element within the *RFT1* promoter, which is another direct target of OsSUF4, and discovered a 7-bp conserved element (5′-CGGAAAT-3′, Nucleotides −989 to −983). EMSA analysis showed that His-OsSUF4N bound to the *RFT1* promoter fragment containing the 7-bp element (Fig. 5d), indicating that 5′-CGGAAAT-3′ within the promoters of *Hd3a* and *RFT1* is the core element for OsSUF4 binding.

To validate that this element is essential for OsSUF4-binding in vivo, a dual-luciferase reporter assay was performed to evaluate the effect of OsSUF4 on wild-type and mutated *Hd3a* and *RFT1* promoters. The promoter regions of *Hd3a* (1885 bp upstream of the TSS to 156 bp downstream of the TSS) and *RFT1* (1777 bp upstream of the TSS to 253 bp downstream of the TSS) containing the 7-bp element were cloned into a luciferase reporter vector fused with the firefly luciferase reporter (Fig. 6a). When the effector construct $P_{Ubi}$::*OsSUF4* and the reporter construct were co-transformed into rice protoplasts, the activity of firefly luciferase driven by wild-type *Hd3a* and *RFT1* promoters was dramatically enhanced to 12-fold and 9-fold,

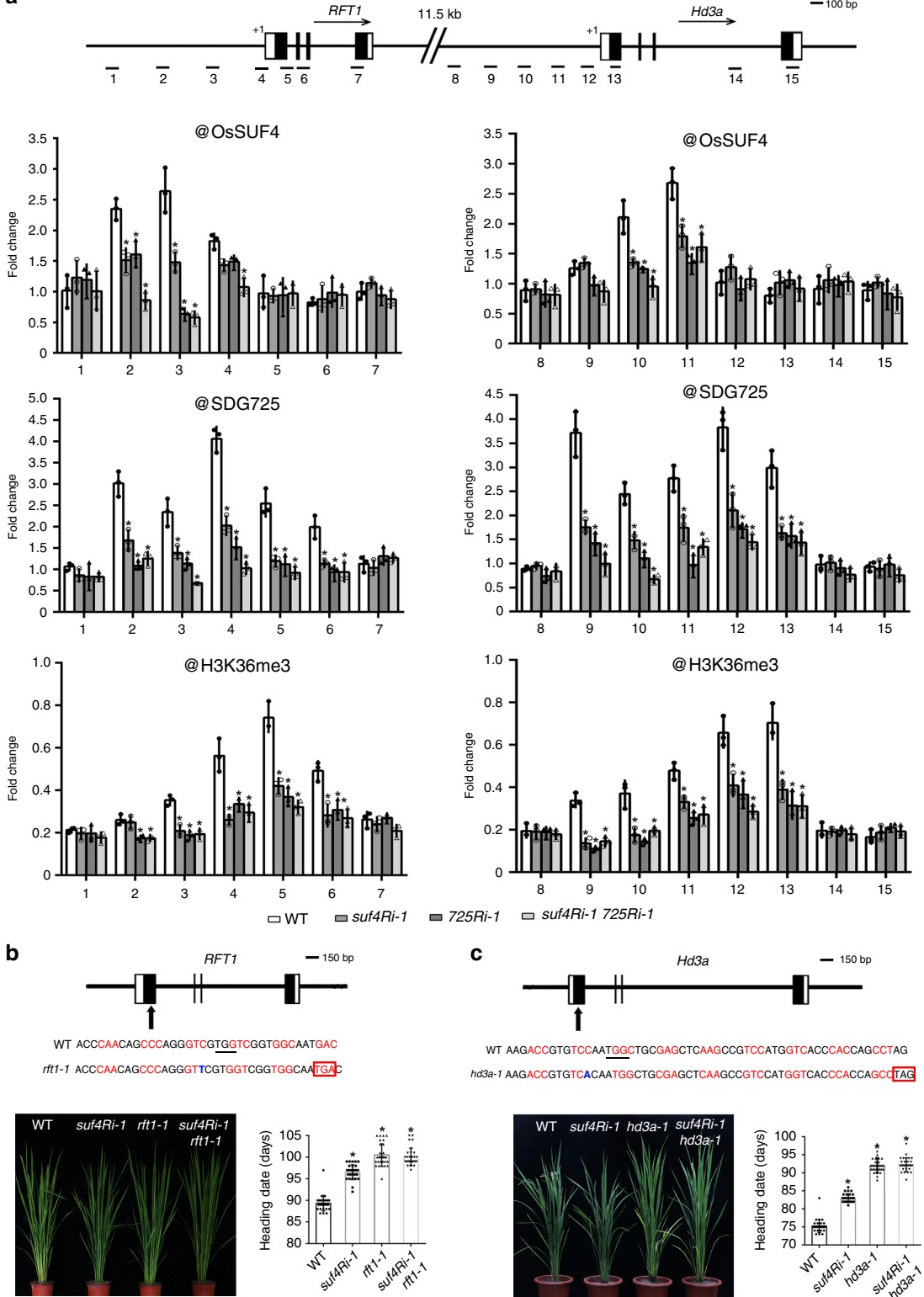

**Fig. 4** OsSUF4 and SDG725 enhance the association of each other at target genes. **a** ChIP analyses at *RFT1* and *Hd3a* promoters using antibodies against OsSUF4, SDG725, and H3K36me3 in WT, *suf4Ri-1*, *725Ri-1*, and *suf4Ri-1 725Ri-1* plants, respectively. Upper panel: schematic representation of *RFT1* and *Hd3a* structures and fragments examined in ChIP-PCR. Values are the mean ± SD of three individual biological replicates normalized to the internal control *OsUbiquitin5*. Asterisks indicate significant differences between indicated genotypes and WT (Student's *t*-test: *$P$ < 0.01). **b, c** CRISPR/Cas9-mediated target mutagenesis of *RFT1* and *Hd3a*, respectively. Upper panel: schematic representation of target site. Nucleotide insertion (shown in blue) resulted in an early stop codon (Red box) for both *RFT1* and *Hd3a*. Left panel: phenotypes of single and double mutants. Right panel: heading date of related plants grown in paddy fields under LD (Shanghai) conditions (**b**) or SD (Sanya) conditions (**c**). Values are the mean ± standard deviation for 30 independent plants. Asterisks indicate significant differences between WT and mutants (Student's *t*-test: *$P$ < 0.01). Source data are provided as a Source Data file

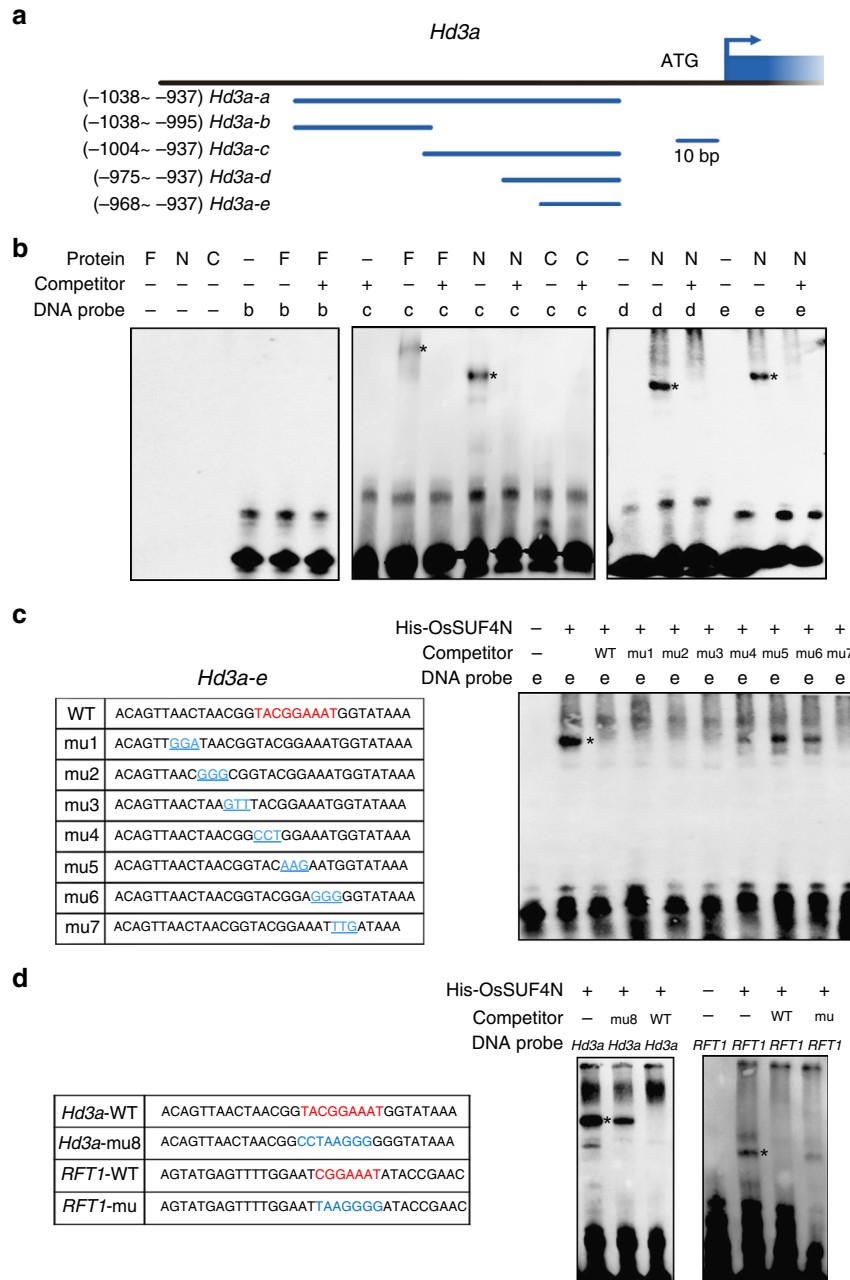

**Fig. 5** OsSUF4 binds to a specific 7-bp DNA element. **a** Schematic representation of the truncated *Hd3a* promoters. **b** Mapping the OsSUF4-binding site within the *Hd3a* promoter. Asterisks indicate the position of the shifted bands. F, N, and C represent the purified full-length, N-terminal, or C-terminal His-OsSUF4, respectively. DNA probe: *Hd3a-b*, *Hd3a-c*, *Hd3a-d*, and *Hd3a-e*. Competitors: unlabeled probes (100-fold excess). **c** Left panel: wild-type (WT) and mutated (mu) DNA probes corresponding to *Hd3a-e* (key element is highlighted in red and mutated bases are shown in blue). **d** EMSA analysis using mutations of the 9-bp element 5′-TACGGAAAT-3′ within *Hd3a-e* and the 7-bp element 5′-CGGAAAT-3′ within *RFT1*. Source data are provided as a Source Data file

respectively, compared with the negative controls. For mutated promoters with deletions of the 7-bp element, firefly luciferase reporter activity was dramatically reduced (Fig. 6a), supporting the fact that this element is essential for *Hd3a* and *RFT1* activation by OsSUF4 in planta.

We then generated transgenic rice expressing β-glucuronidase (GUS) driven by the native promoters of *RFT1/Hd3a* ($P_{Hd3a}$::GUS/$P_{RFT1}$::GUS) and mutated promoters lacking the 7-bp element ($P_{Hd3aΔe}$::GUS/$P_{RFT1Δe}$::GUS). GUS signals in transgenic rice were significantly reduced upon deletion of the 7-bp element within either the *RFT1* or *Hd3a* promoter (Fig. 6b). A histochemical GUS assay in leaf blades again confirmed that the

7-bp element is important for normal expression of *Hd3a* and *RFT1* in rice (Fig. 6c). Taken together, we conclude that OsSUF4 activates the expression of *Hd3a* and *RFT1* by directly binding to their promoters in rice, and that the 7-bp sequence within *RFT1* and *Hd3a* promoters is the core *cis*-element for OsSUF4 recognition.

As the counterparts of SDG725 and OsSUF4 in *Arabidopsis*, the interaction between SDG8 and AtSUF4 was previously reported[34]. AtSUF4 binds to the 5′-CCAAATTTTAAGTTT-3′ element within the promoter of the floral repressor gene *FLOWERING LOCUS C* (*FLC*) in *Arabidopsis*[34,35]. By analyzing the functional substitutability of OsSUF4 and AtSUF4, we found

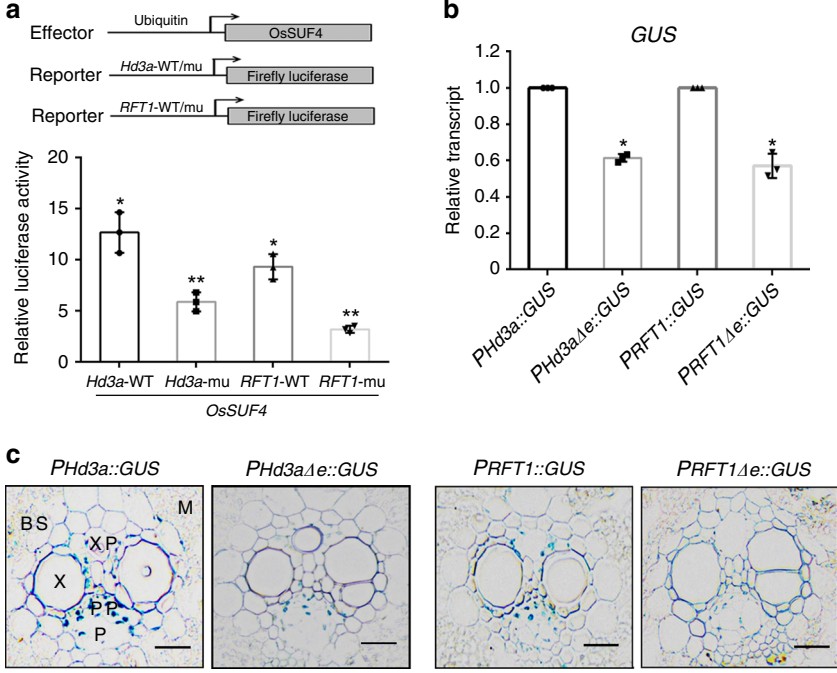

**Fig. 6** The 7-bp DNA element is important for OsSUF4 activation in rice. **a** Dual luciferase assay to test the ability of OsSUF4 to activate wild-type (WT) or mutated *Hd3a* and *RFT1* promoters in planta. Upper panel: schematic representation of effector and reporter plasmids used in the transient transformation assay in rice protoplasts. Relative luciferase activities of each sample refer to the values normalized to their individual controls. The control for each sample was the same individual reporter construct without co-transfection with the effector construct. Values are the mean ± SD of three independent biological replicates. *Significant differences between WT and negative control (Student's *t*-test: *P < 0.01). **Significant differences between WT and mutated reporters (Student's *t*-test: **P < 0.01). **b** Relative transcripts of *GUS* by RT-PCR analysis in $P_{Hd3a}$::GUS, $P_{RFT1}$::GUS, $P_{Hd3a\Delta e}$::GUS, and $P_{RFT1\Delta e}$::GUS plants (Student's *t*-test: *P < 0.01). **c** Transverse leaf section using 35-day-old transgenic rice plants of $P_{Hd3a}$::GUS, $P_{RFT1}$::GUS, $P_{Hd3a\Delta e}$::GUS, and $P_{RFT1\Delta e}$::GUS under SD conditions. M mesophyll cells, BS bundle sheath cells, P phloem, PP phloem parenchyma, V vascular bundle, X xylem, XP xylem parenchyma (Scale bars: 20 μm). Source data are provided as a Source Data file

that OsSUF4 binds to the *Arabidopsis FLC* promoter containing the element: 5′-CCAAATTTTAAGTTT-3′ in vitro, whereas AtSUF4 does not bind the rice *Hd3a-e* fragment (Supplementary Fig. 8a). We then introduced constructs of *AtSUF4* and *OsSUF4* driven by the native promoter and terminator of *AtSUF4* ($P_{AtSUF4}$::*AtSUF4* and $P_{AtSUF4}$::*OsSUF4*) into the *Arabidopsis suf4-2* mutant[27,36] for functional comparison. 15 and 10 transgenic lines expressing *AtSUF4* or *OsSUF4* in suf4-2 background were respectively obtained. The transcript levels of *OsSUF4* in two independent transgenic lines of $P_{AtSUF4}$::*OsSUF4* were similar with that of *AtSUF4* in a line of $P_{AtSUF4}$::*AtSUF4* (Supplementary Fig. 8b). In *suf4-2* mutant, the transcription of *FLC* was completely suppressed, and the *FLC* suppression was rescued by the introduction of *AtSUF4* but not by *OsSUF4* under the control of the *AtSUF4* promoter (Supplementary Fig. 8c), indicating the functional divergence of the two homologs in planta.

**More target genes of OsSUF4 in the rice genome**. We then investigated whether additional genes contained the 7-bp element were recognized by OsSUF4. By screening the rice genome, 3809 genes were discovered to contain the 7-bp element within their promoter regions (Supplementary Data 2). Considering that OsSUF4 acts as a transcription activator and that *OsSUF4* knockdown should down-regulate target genes, we conducted RNA-sequencing to analyze the global profile of the *suf4Ri-1* mutant. A list of 1863 genes had been found down-regulated to more than 1.5-fold in *suf4Ri-1* as compared to the wild-type control (Supplementary Data 3). This list does not include *RFT1*

and *Hd3a*, because these floral genes are expressed at very low levels in young rice plants, such as 14-day-old seedlings used in our RNA-seq analysis. Among the 1863 down-regulated genes, 132 genes contain each at least one 7-bp OsSUF4-binding element (Fig. 7a) and thus represent candidate direct targets of OsSUF4. The other genes might be indirectly down-regulated. Of the 132 genes, 100 genes (75.8%) showed enriched H3K36me3 modification (Fig. 7b, Supplementary Data 4). Among 1731 down-regulated genes not containing the 7-bp element, 1000 sets of 132 randomly selected genes were checked for H3K36me3 enrichment analysis (Supplementary Fig. 9). The numbers of H3K36me3 enriched genes from 1000 sets are shown in histogram as Supplementary Fig. 9. The percentage for the gene number over 100 is 2.6% (26 times among 1000 sets, *P*-value < 0.05), supporting that the 100 genes containing the 7-bp element were significantly enriched of H3K36me3 (Supplementary Fig. 9). Moreover, H3K36me3-enrichment downstream of TSS for these 100 genes was significantly higher as compared to the other H3K36me3-enriched genes (*n* = 16,046) in the wild-type rice genome (Fig. 7c), supporting the idea that OsSUF4 recruits SDG725 to promote H3K36me3 establishment at 5′ ends of target genes.

To further validate the OsSUF4 target genes, we randomly selected seven of the 100 genes containing the 7-bp element within their promoters (Fig. 7a), with enriched H3K36me3 modification (Fig. 7b), and which were down-regulated in the *suf4Ri-1* mutant compared with the wild type (Supplementary Fig. 10). EMSA tested the direct binding of OsSUF4 to target genes in vitro using a 35-bp tandem repetitive sequence consisting of 5 × 7-bp elements as the positive control (Fig. 7d).

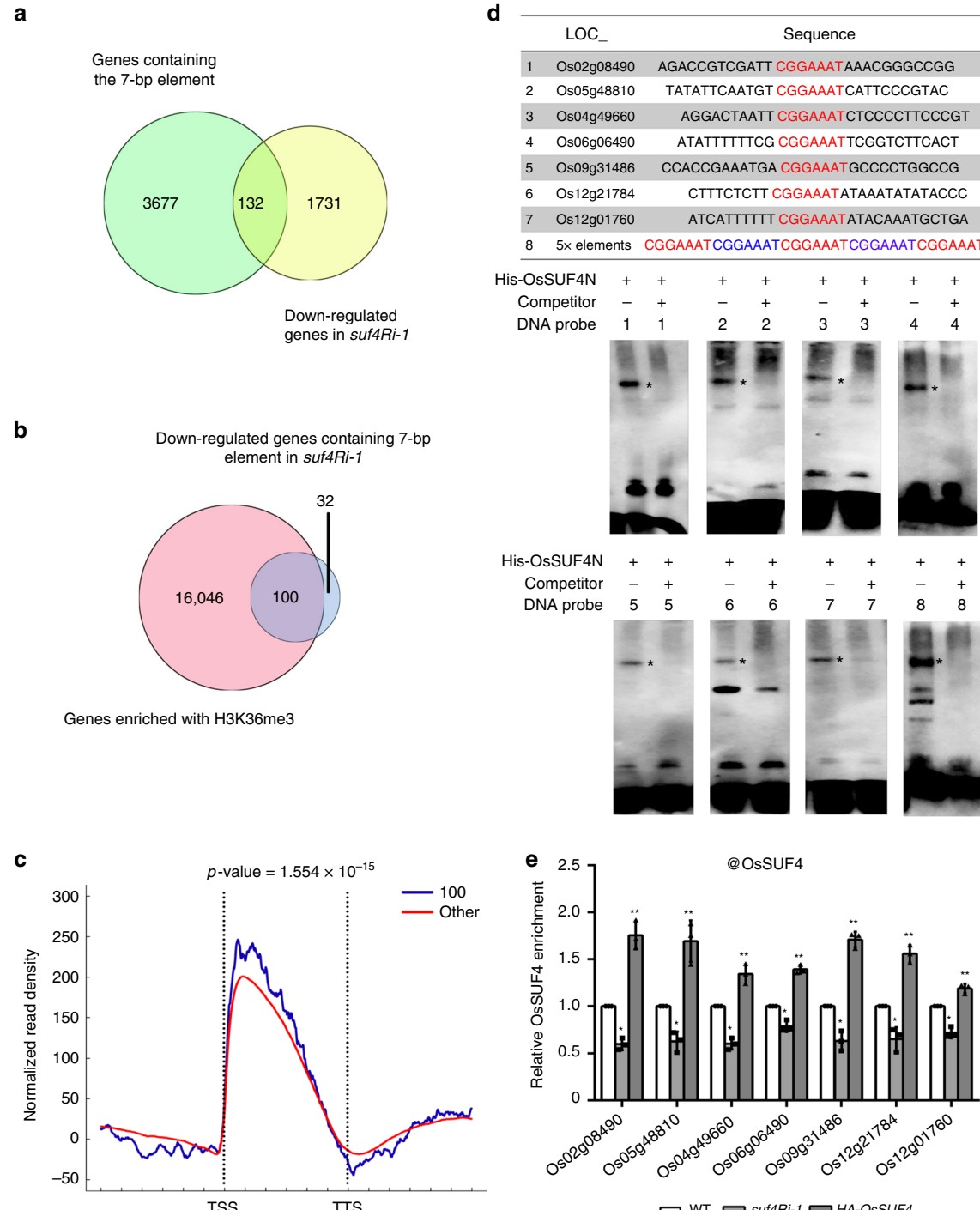

**Fig. 7** OsSUF4 binds to more target genes. **a** Venn diagram showing comparison of total genes containing the 7-bp element in the rice genome (green) and the down-regulated (over 1.5-fold) genes in the *suf4Ri-1* mutant plants (yellow). **b** Venn diagram showing comparison of total genes enriched of H3K36me3 in the wild-type rice genome (rose) with genes containing the 7-bp element and down-regulated in *suf4Ri-1* (blue). **c** H3K36me3 distribution pattern of the 100 genes identified in (**b**) as compared to the other 16,046 H3K36me3-enriched genes in WT rice. The indicated *P*-value was calculated by the Kolmogorov–Smirnov test. **d** EMSA analysis with purified His-OsSUF4N and probes in the upper table. Unlabeled probes were used as competitors (100-fold excess). Asterisks indicate the position of shifted bands. **e** ChIP analysis for OsSUF4 enrichment at the promoter regions of the seven genes using an antibody against OsSUF4 in WT, *suf4Ri-1*, and *P~Ubi~::HA-OsSUF4* plants. Values are the mean ± SD of three individual biological replicates normalized to the internal control *OsUbiquitin5*. *Represents significant differences between WT and *suf4Ri-1* (Student's *t*-test: *P < 0.01). **Represents significant differences between WT and *P~Ubi~::HA-OsSUF4* (Student's *t*-test: **P < 0.01). Source data are provided as a Source Data file

EMSA showed that OsSUF4 directly bound to the promoter fragments of the seven genes, suggesting that the 7-bp element was able to be recognized by OsSUF4 in vitro (Fig. 7d). A ChIP assay was performed to confirm the protein–DNA binding in vivo. As expected, OsSUF4 enrichment at the promoters of the seven genes was significantly decreased in the *suf4Ri-1* mutant and increased in plants overexpressing HA-OsSUF4 (Fig. 7e), indicating that OsSUF4 has more target genes in the rice genome.

Taken together, our results suggest that in addition to the key flowering genes *RFT1* and *Hd3a*, OsSUF4 binds to a set of target genes whose promoters contain a 7-bp core element, and may contribute to H3K36me3 establishment at the 5′ end of gene body by interacting with the H3K36-specific methyltransferase SDG725.

**Structural basis for target DNA recognition by OsSUF4.** To unravel the structural basis underlying target DNA recognition by OsSUF4, we carried out a crystallographic study of the zinc finger domain (aa 10–100) of OsSUF4. The structure was solved by a Se-SAD phasing method and refined up to 1.95 Å resolution (Supplementary Table 1). As depicted in Fig. 8a, the OsSUF4 zinc finger domain contained two $C_2H_2$ type zinc finger motifs with different folding. The first zinc finger motif adopts a canonical $C_2H_2$ fold containing a short antiparallel β-sheet followed by one α-helix; instead of a β-sheet, the second zinc finger motif contains a coiled-coil (Fig. 8a, Supplementary Fig. 11). The zinc-

coordinating cysteines and histidines located in the β-sheet/ coiled-coil and the α-helix, respectively, and coordination of the two zinc ions are very similar (Supplementary Fig. 12a). In many known $C_2H_2$-type zinc finger proteins, neighboring $C_2H_2$ motifs are connected by long and flexible loops; however, only two residues were shown to be embedded in the junction region of the two zinc finger motifs of OsSUF4, preventing them from synchronously nestling into the DNA major groove. Structural comparison revealed that folding of the first zinc finger motif of OsSUF4 is similar to that of the Tramtrack protein (PDB_ID: 2DRP) zinc finger DNA-binding domain[37]. However, different from Tramtrack, the second zinc finger motif of OsSUF4 floats over the DNA, suggesting that the first motif is probably responsible for DNA binding by OsSUF4 (Supplementary Fig. 12b).

Notably, the first zinc finger domain is enriched with positively charged residues (R18, K24, K31, K33, K36, and K43), forming an electropositive interface (Fig. 8b, Supplementary Fig. 12c). To test

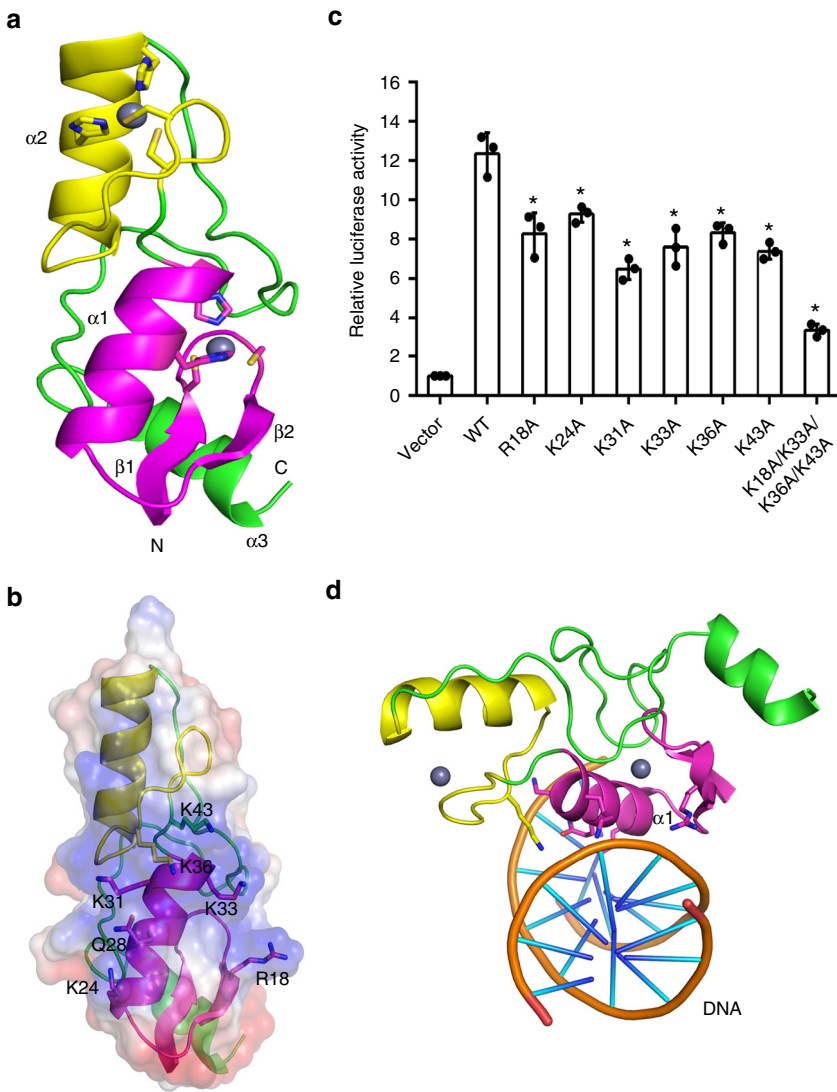

**Fig. 8** Identification of residues crucial to the function of OsSUF4. **a** Overall structure of OsSUF4. The first and second zinc finger domains are shown in magenta and yellow, respectively. Zinc ions are shown as gray spheres. **b** The detailed orientations of R18, K24, K31, K33, K36, and K43 within the first zinc finger domain of OsSUF4. **c** Dual luciferase assay showing *Hd3a* promoter activation by wild-type or mutated OsSUF4. Relative luciferase activity refers to normalization of firefly luciferase activity to the renilla luciferase activity as the internal control, then normalization to the vector (set as 1). Values are the mean ± SD of three independent biological replicates. Asterisks indicate significant differences between WT and mutants (Student's *t*-test: *$P < 0.01$). **d** The proposed OsSUF4–DNA-binding model. The first zinc finger domain of OsSUF4 is proposed to contact DNA via positively charged residues from the major groove side. Source data are provided as a Source Data file

whether these residues were involved in DNA binding by OsSUF4, we performed mutagenesis and isothermal titration calorimetry analysis. As shown in Supplementary Fig. 12d, the wild-type OsSUF4 zinc finger domain bound to the substrate DNA (5′-GGGTACGGAAATGGTA-3′) with a $K_d$ value of 51.28 ± 3.68 μM, whereas R18A, K24A, K31A, K33A, K36A, K43A, and R18A/K33A/K36A/K43A mutants all lost their DNA-binding ability. To further confirm the functional importance of these residues, we performed a dual luciferase assay in planta (Fig. 8c). Compared with wild-type OsSUF4, Hd3a promoter activation by all single mutant proteins was significantly reduced; a more dramatic reduction was observed for the R18A/K33A/K36A/K43A mutant. Taken together, our findings suggested that the positive residues within the surface of the first zinc finger motif are essential for target DNA binding by OsSUF4 (Fig. 8d).

## Discussion
In this study, using the major H3K36 methyltransferase SDG725 as bait, we identified OsSUF4 as a $C_2H_2$-type zinc finger transcription factor in rice that interacts with SDG725 in vitro and in vivo. Several lines of evidence indicated that via the interaction with OsSUF4, SDG725 is recruited to the promoters of florigen RFT1 and Hd3a for H3K36me3 deposition to promote gene activation and rice plant flowering. We also mapped a 7-bp DNA element, 5′-CGGAAAT-3′, as the core element for OsSUF4 binding. Deletion of this 7-bp cis-element clearly reduced the promoter activity of RFT1 and Hd3a. A further 3809 genes were shown to contain the 7-bp cis-element within their promoter regions throughout the rice genome, of which 132 were down-regulated over 1.5-fold in the OsSUF4 knockdown mutant. Among the 132 genes containing the 7-bp element and down-regulated in the OsSUF4 knockdown mutant, 100 showed enriched H3K36me3 modification. Seven randomly selected genes of these 100 were all direct targets of OsSUF4. Collectively, our results help to explain H3K36me3 at the 5′ ends of RFT1 and Hd3a and may contribute to a better understanding of the situation genome-wide.

OsSUF4 homologs are widespread in eukaryotes but absent in yeast (Supplementary Fig. 1). ZNF207 and SETD2 were possible counterparts of OsSUF4 and SDG725 in humans; however, no interaction between SETD2 and ZNF207 was detected in a yeast two-hybrid assay (Supplementary Fig. 13). As the counterpart of OsSUF4 in Arabidopsis, AtSUF4 was previously reported to interact with the Arabidopsis H3K36 methyltransferase SDG8[34], while the DNA-binding element and target genes of AtSUF4 seem to largely differ from those of OsSUF4, implying that during evolution, dicot and monocot plants evolved functionally separate SUF4 proteins. Consistently, our rescue experiments showed that AtSUF4 but not OsSUF4 can rescue the Arabidopsis suf4-2 mutant (Supplementary Fig. 8c). Nevertheless, the methylation deposition mechanism mediated by the SUF4 transcription factor and H3K36 methyltransferase seems to be conserved in Arabidopsis and rice. Thus, we propose that OsSUF4/AtSUF4 help SDG725/SDG8 to target the 5′ end of gene body regions and promote H3K36 methylation, which is in line with the apparent TSS-proximal pattern of H3K36me3 in plants. The interaction between enzymes and transcription factors might influence the targeting of the enzymes and the distribution of the corresponding modifications. SDG725, by interaction with the transcription factor OsSUF4, is enriched close to the TTS regions of some OsSUF4-targeted genes, such as RFT1 and Hd3a. SDG725 deficiency also impairs OsSUF4 binding at the targets, suggesting a reciprocal enhancing mechanism. The C-terminal domain of OsSUF4 interacts with SDG725 (Fig. 1d) and the N-terminal zinc finger domain of

OsSUF4 is responsible for DNA binding (Fig. 8). A speculation would be that the complex formation with SDG725 might change the structure of OsSUF4 for a better association with DNA. SDG725 contains a CW-domain capable of binding with H3K4me1[38], indicating that SDG725 may interact with chromatin also through OsSUF4-independent mechanism. Finally, SDG8 and AtSUF4 were found in larger complex containing other protein components[34]. A similar complex may also exist in rice, which acts as a functional unit in H3K36me3 deposition and active transcription of the florigens RFT1 and Hd3a, as well as possibly other OsSUF4-target genes.

Several pieces of evidence suggest that the link between transcription factors and epigenetic modifiers is not limited to H3K36 methylation. In Arabidopsis, several transcription factors were found to recruit Polycomb repressive complex 2 for H3K27me3-mediated gene silencing, such as telomere repeat-binding factors[39]. Moreover, the H3K4me3 demethylase AtJMJ14 interacted with transcription factors NAC050 and NAC052 to bind the CTTG(N)5CAAG motif and regulate various biological processes[40–49]. In rice, the $C_2H_2$ zinc finger protein SDG723/OsTrx1/OsSET33 Interaction Protein 1 was recently reported to interact with OsTRX1, which is responsible for establishing H3K4me3 on flowering gene Ehd1 and promoting flowering[50]. Further exploration into trans-acting factors and cis-acting DNA elements involved in chromatin epigenetic regulation will help our understanding of methylation establishment and spreading, and their related biological significance.

## Methods
**Phylogenetic analysis.** Alignment of amino acid sequences of OsSUF4 homologs in different species was performed via the ClustalW program[51]. The resulting file was subjected to phylogenetic analysis using MEGA4.0 software[52]. The trees were constructed with the following settings: Neighbor-Joining for tree inference, complete deletion option for each class analysis, and bootstrap test of 1000 replicates for internal branch reliability.

**ChIP-seq analysis.** Raw published H3K36me3 ChIP-seq data for different species were downloaded from the Sequence Read Archive (SRA) database (https://trace.ncbi.nlm.nih.gov/Traces/sra/sra/). Access numbers are SRP048526 for yeast[53], SRP007859 for C. elegans (http://www.modencode.org), SRP023380 and SRP023365 for D. melanogaster (http://www.modencode.org), SRP016121 for mice[54], SRP132532 for humans[55], SRP002100 for Arabidopsis[56], and SRP063912 for rice[8]. SRA files for all species except Arabidopsis were transformed to FASTQ files using the fastq-dump package in SRAToolkit v2.9.0[57]. Bowtie v1.9 was used to map H3K36me3 ChIP-seq data to the reference genome of each species. Arabidopsis H3K36me3 SRA data were transformed to FASTQ files with the abi-dump package in SRAToolkit v2.9.0, then mapped to the Arabidopsis genome (TAIR10) with SHRiMP v2.2.3 software[58]. Unique (MAPQ > 20) and non-redundant mapped reads were generated with SAMtools v1.9[59], and aggregated plots and heatmaps were constructed with deepTools v2.0 and R v5.0 software.

For the distribution analysis of H3K36me3 on different gene groups, regions from 3 kb upstream of the TSS site to 3 kb downstream of the TTS site were plotted using deepTools v2.0[60]. Briefly, gene body regions of different length genes were fitted to 1 kb and divided into 300 bins; regions of 3 kb upstream and downstream of the TTS were, respectively, divided into 300 bins. The methylation levels of each bin were calculated by the reads per kilobase of transcript per million mapped reads value and calibrated to the input. The aggregated plots were treated with R v5.0 software. Visualization of ChIP-seq profiles was performed in the Integrative Genomics Viewer v2.3.46[59] with deepTools v2.0.

To identify H3K36me3-enriched genes in WT, SICER.sh from SICER v1.1 software[61] was used to identify the H3K36me3-enrichment region (peaks) by comparing the ChIP-seq library with the input DNA library (parameters: W-200, G = 200, FDR < 1e−3 for H3K36me3). Significant peaks were found with FDR < 1e−3 and IP-DNA/Input-DNA ≥ 2. ChIPpeakAnno[62] from http://www.bioconductor.org/ was performed for peak annotation. Genes (1 kb upstream of TSS and gene body regions) containing significant H3K36me3 peaks were considered H3K36-enriched genes. The P-value was calculated by the Kolmogorov–Smirnov test.

**Plant transformation and growth conditions.** To create RNAi constructs, DNA fragments containing nucleotides from 23 to 359 and 363 to 612 of the OsSUF4 open-reading frame were selected to produce the hairpin structure using primers listed in Supplementary Data 1. The hairpin structures were then cloned into the

pDS 1301 vector[63], resulting in $P_{35S}$::OsSUF4-RNAi-1 and $P_{35S}$::OsSUF4-RNAi-2. The full-length cDNA of OsSUF4 was amplified by RT-PCR using primers listed in Supplementary Data 1. The resulting product fused with DNA encoding HA-tag was then cloned into the pU1301 plant expression vector containing the maize (Zea mays) Ubiquitin promoter, yielding $P_{UBi}$::HA-OsSUF4. Regarding CRISPR/Cas9 plants, single guide RNA oligos were inserted into the BGK032 vector (BGK032, BIOGLE GeneTech, Hangzhou, China).

To generate transgenic constructs carrying $P_{RFT1}$::GUS, $P_{Hd3a}$::GUS, $P_{RFT1\Delta e}$::GUS, and $P_{Hd3a\Delta e}$::GUS, wild-type or mutated promoters of RFT1 and Hd3a were inserted into the pCAMBIA1391Z vector. The primers for creating these constructs are listed in Supplementary Data 1. Agrobacterium tumefaciens (strain EHA105)-mediated transformation to Oryza sativa spp. Japonica cv Nipponbare (http://signal.salk.edu/cgi-bin/RiceGE) was performed according to a previously described procedure. Briefly, after being induced, the calli were immersed in the Agrobacterium tumefaciens suspension for 5 min, and then transformed to a J3 medium (L3, 2.5 mg l$^{-1}$, 2,4-D, 500 mg l$^{-1}$ proline, 500 mg l$^{-1}$ glutamine, 3% maltose, 0.25% phytagel, 200 mM acetosyringone, 1% glucose, pH 5.8). Differentiation of resistant callus was on DL3 medium[64]. Plants were grown in two locations, namely Shanghai and Sanya, which represent natural LD and SD conditions, respectively. Seedlings used for quantitative RT-PCR and ChIP assays in rice were cultured in artificial growth chambers under LD conditions (14 h 30 °C: 10 h 28 °C, light:dark) or SD conditions (10 h 30 °C: 14 h 28 °C, light:dark).

The Arabidopsis mutant suf4-2 (SALK_093449) was obtained from the Arabidopsis Biological Resource Center (ABRC, http://www.arabidopsis.org) and the Saskatoon collection[65], and the mutant has been previously described[28]. To generate $P_{AtSUF4}$::HA-OsSUF4 and $P_{AtSUF4}$::HA-AtSUF4 transgenic plants, a DNA fragment including the promoter (1275 bp upstream of the TSS) and terminator (425 bp downstream of the TTS) of AtSUF4, the full length coding sequence (CDS) encoding OsSUF4 and AtSUF4 were amplified and inserted into the pCAMBIA1300 vector. The resulting construct was transformed into the Arabidopsis suf4-2 mutant. Arabidopsis plants were grown under a photoperiod of 16 h light and 8 h dark.

**Yeast two-hybrid assay.** Total RNA for cloning the cDNA library into pGAD vectors was extracted from wild-type plants using Matchmaker Library Construction and Screening Kits (Clontech, Shiga, Japan). Yeast screening was performed using truncated SDG725 protein (SDG725C, amino acids 1240–2150) as the bait by yeast mating. Full-length or truncated CDS of OsSUF4, SDG725, and ZNF207 were amplified and cloned into pGADT7 or pGBKT7 (Clontech) using the primers listed in Supplementary Data 1, resulting in constructs pGADT7-OsSUF4, pGADT7-OsSUF4N, pGADT7-OsSUF4C, pGADT7-ZNF207, pGBKT7–725C, and pGBKT7-SETD2. The yeast two-hybrid assay was performed according to the manufacturer's protocol (Clontech) and the interaction was screened on media lacking tryptophan, leucine, and adenine (SD −W/−L/−A).

**Pulldown assay.** Full-length or truncated cDNA of OsSUF4 and AtSUF4 (OsSUF4, OsSUF4N, OsSUF4C, and AtSUF4) and C-terminal cDNA of SDG725 (SDG725C) were amplified using primers listed in Supplementary Data 1, then cloned into the pET32a expression vector (Novagen, Madison, WI, USA) and pGEX-4T1 (GE Healthcare, Milwaukee, WI, USA), resulting in His-OsSUF4, His-OsSUF4N, His-OsSUF4C, His-AtSUF4, and GST-SDG725C. Protein expression analysis and purification for the pulldown assay in vitro were performed using a kit according to manufacturer's recommendations (GE Healthcare). For the pulldown assay using over-expressed HA-OsSUF4 plants, total nuclear extracts were incubated with GST or GST-SDG725C beads in pulldown buffer (50 mM Tris pH 7.5, 100 mM NaCl, 1 mM ethylenediaminetetraacetic acid [EDTA], 10% glycerol) for 2 h. Pulldown fractions were analyzed by western blotting using an anti-GST antibody (SG4110-01, Shanghai Genomics, Shanghai, China) at a 1:1000 dilution, anti-His antibody (SG4110-38, Shanghai Genomics) at a 1:1000 dilution, and anti-HA antibody (ab9110, Abcam, Cambridge, MA, USA) at a 1:1000 dilution.

**BiFC assay.** OsSUF4 cDNA was amplified using primers listed in Supplementary Data 1 and cloned into pXY103 and pXY106 vectors. SDG725C cDNA was amplified by primers listed in Supplementary Data 1 and cloned into the pXY104 vector. Then, different groups of constructs were transiently expressed in the leaves of 4–8-week-old Nicotiana benthamiana plants via agroinfiltration. The fluorescence was observed 2 days after infiltration using a confocal laser scanning microscope (LSM 710, ZEISS, Germany).

**Gene transcription analysis.** To analyze expression levels of flowering-related genes, 30-day-old (SD) and 35-day-old (LD) rice shoots were harvested for each sample at indicated time points for total RNA extraction using TRIzol reagent according to the manufacturer's instructions (15596018, Invitrogen, Carlsbad, CA, USA). For RNA-seq analysis, 14-day-old seedlings were collected under SD conditions. Reverse transcription was performed using Improm-II reverse transcriptase (A3801, Promega, Madison, WI, USA) according to the manufacturer's protocol. Quantitative PCR was performed using gene-specific primers listed in Supplementary Data 1. Rice OsUbiquitin5 and Arabidopsis ACTIN2 were used as reference genes to normalize the data.

**ChIP–PCR assay.** The ChIP assay was performed using 14-day-old or 30-day-old (SD) rice seedlings. Briefly, 2.0 g seedlings were harvested and fixed using fixation buffer (0.4 M sucrose, 10 mM Tris–HCl pH 8.0, 1 mM EDTA, 1% formaldehyde, and 1 mM PMSF). The nucleus were extracted using lysis buffer (50 mM HEPES pH 7.5, 150 mM NaCl, 1 mM EDTA, 1% Triton X-100, 5 mM β-mercaptoethano, 10% glycerol, and Protease Inhibitor cocktail). Then, DNA was sonicated into fragment below 500-bp using lysis buffer with 0.8% SDS, followed by immuno-precipitation with anti-H3K36me3 (ab9050, Abcam) at a 1:600 dilution, anti-HA (ab9110, Abcam) at a 1:300 dilution, anti-SDG725 at a 1:300 dilution and anti-H3 (ab1791, Abcam) at a 1:300 dilution[66]. The monoclonal antibody against OsSUF4 was produced by Abmart using the peptide DVLAAHYGEE (1:300 dilution). After gradual washes using low-salt buffer (50 mM HEPES pH 7.5, 1 mM EDTA pH 8.0, 150 mM NaCl), high-salt buffer (50 mM HEPES pH 7.5, 1 mM EDTA pH 8.0, 500 mM NaCl), LiCl wash buffer (10 mM Tris–HCl pH 8.0, 1 mM EDTA pH 8.0, 0.5% NP-40, 0.25 M LiCl) and TE buffer (10 mM Tris–HCl pH 8.0, 1 mM EDTA pH 8.0). The complex was eluted using elution buffer (1% SDS, 0.1 M NaHCO$_3$). The precipitated DNA was obtained by reversing the crossing-linking of protein–DNA. To determine the enrichment of precipitated DNA, quantitative PCR was performed using TB Green™ Premix Ex Taq™ II kit (RR8020A, Takara, Shiga, Japan) and specific primers listed in Supplementary Data 1.

**Electrophoretic mobility shift assay.** EMSA was performed using a LightShift Chemiluminescent EMSA KIT (20148, Thermo Fisher Scientific, Waltham, USA). Briefly, each reaction contained 5 μg purified protein, 0.02 μM labeled probe, and 100-fold unlabeled probe, 2 μL binding buffer, and 0.5 μL poly(dI-dC). Binding reactions were incubated at 25 °C for 30 min. Then the reaction mixture was subjected to electrophoresis at 100 V for 2 h using 5% polyacrylamide gel. After transferred to a positive charged nylon membrane (INYC00010, Millipore, MA, USA), the membrane was cross-linked below ultraviolet lamp. The membrane was treated with the LightShift Chemiluminescent EMSA Kit and then was exposed to film.

**Protoplast transformation and dual-luciferase reporter assay.** Rice protoplast isolation and transformation were performed using 2-week-old seedlings. Briefly, different promoters including wild-type and mutated forms were amplified and inserted into the pGreenII-0800-LUC vector using gene-specific primers listed in Supplementary Data 1. Wild-type or mutated CDS of OsSUF4 fused with a HA-tag were cloned into the pU1301 vector and used as effectors. Levels from 14-day wild-type seedlings were excised and incubated in enzyme buffer A (1.5% cellulose RS, 0.3% macerozyme, 0.1% pectolyase, 0.6 M mannitol, 10 mM MES, 1 mM CaCl$_2$, 0.1% bovine serum albumin, pH 5.7) for 5 h. After adding an equal volume of W5 buffer (154 mM sodium chloride, 125 mM CaCl$_2$, 5 mM KCl, 2 mM MES, pH 5.7), the harvesting cells were resuspended in MMG buffer (0.6 M mannitol, 15 mM MgCl$_2$, 4 mM MES, pH 5.7). 10 mg of each vector was added to the protoplasts for 10 min using 40% PEG buffer (0.6 M mannitol, 15 mM MgCl$_2$, 4 mM MES, pH 5.7) and then were washed for two times using W5 buffer. At last, the protoplasts were resuspended in incubation buffer (0.6 M mannitol, 4 mM MES, 4 mM KCl, pH 5.7). After incubating for 12–16 h, the protoplasts were spun down and incubated with lysis buffer. Firefly and Renilla luciferase were quantified using a dual-luciferase assay kit (E1910, Promega).

**Histochemical GUS staining.** Five-week-old PRFT1::GUS/PRFT1Δe::GUS (LD) and PHd3a::GUS/PHd3aΔe::GUS (SD) plants were used for GUS staining. Leaf blades were collected and infiltrated with staining solution (50 mM sodium phosphate buffer, pH 7.0, 0.5 mM potassium ferrocyanide, 0.5 mM potassium ferricyanide, 0.1% Triton X-100, 10 mM EDTA, 1% dimethyl sulfoxide, and 0.5 mg/mL X-Gluc) in a vacuum chamber, then incubated at 37 °C for 12 h. Chlorophylls were removed by incubating in 70% ethanol at 65 °C and dehydrating through an ethanol series (50%, 70%, 90%, and 100%) and ethanol/histoclear series. The leaves were embedded in paraffin and sectioned at a thickness of 10 μm using an Ultracut UCT ultramicrotome (Leica).

**RNA-seq analysis.** Library construction and sequencing were performed as previously described[8]. The original paired-end reads were first trimmed using CUTADAPT v1.10. Then, TOPHAT2 v2.0.13[69] was used to align adapter-removed raw reads to the complete reference genome of Nipponbare (japonica) rice (http://rice.plantbiology.msu.edu). Normalized gene transcription levels were calculated as fragments per kilobase of exon per million fragments mapped by CUFFLINKS v2.2.1[67]. Next, differentially expressed genes (DEGs) were identified using the CUFFDIFF program, which is a subpackage of CUFFLINKS. DEG identification was performed at a threshold of a more than 1.5 fold-change and an adjusted P-value < 0.05.

BEDToolsv2.17.0 and "Biostrings" package were used to search for the promoter (2 kb upstream of the TSS) containing the "CGGAAAT" element. Genes down-regulated to more than 1.5-fold were referred to as down-regulated genes in suf4Ri-1. Venn diagrams were generated using 'Venn Diagram' packages in R software.

**Protein crystallization and structure determination.** The DNA fragment encoding the OsSUF4 zinc finger domain (amino acids 10–100) was subcloned into the pET28-SUMO vector and expressed in *Escherichia coli* BL21(DE3) competent cells. To facilitate structural determination, Se-Met-substituted OsSUF4 was also prepared. Proteins were purified by the Ni-chelating column and size-exclusion chromatography (Superdex G75, GE Healthcare), stored in 20 mM Tris–HCl pH 8.0, 100 mM NaCl, 2 mM dithiothreitol buffer, and concentrated to 20 mg/mL prior to use. All crystals were grown at 18 °C using the sitting drop vapor diffusion method. The drop contained equal volumes of protein and reservoir solution (0.1 M Hepes, pH 7.0, 30% w/v PEG6000). X-ray diffraction data were collected on beamline BL17U1 at the Shanghai Synchrotron Radiation Facility. Data processing was carried out using the HKL2000 program[68]. The structure was solved using single-wavelength anomalous diffraction with the Autosol program embedded in the Phenix suite[69]. Structural refinement was performed using the Refmac program of CCP4i[70]. Coot[71] were used for model building. Details of data collection and refinement statistics are summarized in Supplementary Table 1.

**ITC assay.** All mutant proteins used in ITC assays were purified following the same method as wild-type proteins. Oligonucleotides were purchased from Sangon Biotech. To obtain stable and uniform DNA duplexes, complementary oligonucleotides were mixed in a molar ratio of 1:1, heat-denatured at 95 °C, then slowly cooled to 12 °C. ITC was carried out by MicroCal iTC200 (GE Healthcare) in 100 mM NaCl, 20 mM Tris–HCl, pH 8.0 buffer at 25 °C. Then, 2 mM OsSUF4 was titrated into 0.1 mM DNA and the mutants. The thermogram was processed by the ITC data analysis module of Origin 7.0 (MicroCal) and fitted into the one-site-binding mode.

**Reporting summary.** Further information on research design is available in the Nature Research Reporting Summary linked to this article.

## Data availability

RNA-seq data in this study have been deposited in the Sequence Read Archive (SRA) database and are available with the following accession code: SRP186556. PDB accession code of the OsSUF4 zinc finger domain is 6J0D. The data that support the findings of this study are available from the corresponding authors upon reasonable request. The source data of the gels and immunoblots in Figs. 2b, 5b–d, 7d, Supplementary Fig. 2, Supplementary Fig. 5, and Supplementary Fig. 8, as well as the source data underlying Figs. 3b, c, e–g, 4, 6a, b, 7e, 8c, Supplementary Fig. 3, Supplementary Fig. 4, Supplementary Fig. 6, Supplementary Fig. 7, Supplementary Fig. 8b, c, and Supplementary Fig. 10b are provided in the Source Data file.

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

## Acknowledgements

We thank Dr. Jianxiang Liu for pGreenII-0800-LUC vector, and the staff of beamlines BL17U1, BL18U1, and BL19U1 at the Shanghai Synchrotron Radiation Facility for assistance during data collection. This work was supported by the National Natural Science Foundation of China (grants NSFC31570315, 31771420, 31800207, 31571319, and 91519308) and the National Basic Research Program of China (2012CB910500). This research was conducted within the context of the International Associated Laboratory on Plant Epigenome Research (LIA PER).

## Author contributions

A.D. conceived and designed the research. Y.Y. and J.G. supervised the experiments. B.L., Y.L., B.W., Q.L. and J.S. performed the experiments. B.L., Y.L., B.W., Q.L. and J.G. analyzed the data. B.L., B.W., J.G., W.-H.S., Y.Y. and A.D. wrote the manuscript. All authors read, revised, and approved the manuscript.

## Additional information

**Competing interests:** The authors declare no competing interests.

