## [Peer Review File · Nature Communications]

Reviewers' comments:

Reviewer #1 (Remarks to the Author):

The paper by Liu et al. report on the studies on targeting of H3K36me3 methyltransferase SDG725 by OsSUF4. After a long time studies of the epigenetic modification enzymes, the targeting mechanism becomes a hot topic recently. The global distribution of these enzymes to certain chromatin region precisely regulates the function of these enzymes. In this paper, the authors analyzed the global distribution of H3K36me3 from different species and found that plants possess a specific H3K36me3 pattern other than animals. Further, they identified the transcription factor OsSUF4 can target SDG725 to a 7-bp DNA element, shaping the global pattern of plant H3K36me3. The structural studies of the ZF of OsSUF4 potentially identified the DNA binding site. Generally, the paper is well ordered and of good quality. Some issues below can be considered to strengthen the paper.

1. As the author only provided limited information, I do not quite understand the Fig 2b. I assume the author want to show the purification of GST-725C in the left panel which is stained by GST antibody and the pull-down of HA-OsSUF4 which is stained by HA antibody. The authors need to provide more details in the figure caption. Because the molecular sizes of GST-725C and HA-OsSUF4 are quite similar as indicated by the gel. The author need to provide the validation for the antibodies that HA antibody does not react with GST-725C and GST antibody does not react with HA-OsSUF4 by simply staining the raw proteins with the antibodies.
2. Fig 2c caption, more details. What is DIC, YN, and YC? Please provide the full name.
3. Fig 2d. On the top, it should be 'C2H2 type zinc finger domains'.
4. Line163, briefly discuss about why Ghd7 and Hd1 were up regulated in suf4Ri-1 and 725Ri-1.
5. Fig 7a. Among the 1863 gene down regulated by suf4Ri-1, only 132 contain the 7-bp element. The ratio is quite low. Does it contain the discussed key flowering genes, RFT1 and Hd3a? Please discuss.
6. Line 312, the binding affinity is different from the one in fig s7d.
7. Line 560-6-562, supplement the reference for HKL2000, Phenix and Refmac. Which program for model building, Coot or some others? Please add it. Line561, Phenix but not Phoenix.

Reviewer #2 (Remarks to the Author):

In this manuscript a physical interaction was found between the rice SUF4 protein and SDG725 and various and extensive molecular and genetic experiments have been performed to decipher the putative molecular function of this protein complex. Based on the obtained results, the authors postulate that SUF4 targets SDG725 to promoter regions of a specific sub-set of genes in the rice genome, including the key flowering time regulators RFT1 and Hd3a. As a consequence, the H3K36me3 mark is supposed to be deposited at these loci resulting in enhanced expression of these selected genes. An interaction between SUF4-like proteins and SDG proteins, and targeting of SDG proteins to specific loci by this mechanism, has been shown previously in other species. Hence, this finding is not completely new. Furthermore, there are some inconsistencies and points of attention in the presented data, as summarized below:

1. The title: 'The transcription factor OsSUF4 interacts with SDG725 and promotes H3K36me3 establishment towards gene promoter regions' is not clearly reflecting the findings of the paper. H3K36me3 deposition is found in general towards the 5'end of the gene body and not in promoter regions. See also my more detailed comments on this aspect below.
2. For the various ChIP-PCR enrichment analyses three adjacent promoter regions have been selected in RFT1 and Hd3a loci (Fig. 4A). (1) Please indicate the reasoning for selecting these specific region in the proximal promoter region? Theoretically, SUF4 could also bind somewhere else in the locus, or not? (2) The exact resolution of this analysis is not clear from the schematic representation. Please indicate in figure 4 the distance between the three selected promoter

regions and the distance in relation to the TSS. (3) I miss local controls to exclude the possibility that the observed enrichments for the three promoter fragments are not due to specific binding by the Ab-targeted proteins, but e.g. due to a general increased accessibility of these promoter regions because of altered expression levels of these two loci in the mutant backgrounds. Please include a fragment in e.g. an intron or exon (towards the 3' end of the gene) as additional controls.

(4) H3K36me3 enrichment analysis is also done for this promoter region upstream of the TSS. According to the analysis presented in Fig. 1, genome-wide, this region is not enriched or even underrepresented in H3K36me3 deposition. Why was this specific region selected for the H3K36-trimethylation analysis in these two loci? (5) Note that for H3K36me3 enrichment, a comparison has to be made to an IP using a general H3 antibody, to ensure that observed differences are not just reflecting general differences in 'Histone3/chromatin' occupancy of the investigated genomic regions in particular genetic backgrounds.

3. Page 10, line 198-202: 'To further investigate the genetic relationship between OsSUF4 and RFT1/Hd3a, mutants containing a 1-bp thymine insertion within the coding region of RFT1 (known as *rft1*) and a 1-bp adenine insertion within that of Hd3a (*hd3a*), which respectively resulted in an early termination, were generated by clustered regularly interspaced short palindromic repeats (CRISPR)/ CRISPR-associated protein (Cas)9.'. This reads as that an approach was followed to specifically create these two single nucleotide insertions. If so, please describe how this was done? Or did the authors perform a classical CRISPR-based mutagenesis and selected these two specific lines for further analyses? Please describe in more detail how these two mutant lines were generated and identified.

4. EMSA experiments (Fig. 5). I find the results presented in figure 5b puzzling and not convincing because of a lot of background and overexposure, making it difficult to compare exact patterns in the various lanes on the gel. How do the authors explain e.g. that the smaller protein domain (OsSUF4N) results in a larger shift in combination with DNA probe 'c' than the full length OsSUF4 protein? (when I check the pattern for the free 'c' probe in the two gels, the running conditions seem to be more or less the same and hence, this cannot be the cause of the observed large difference in shift). In contrast, the data presented in 5c are clear and providing evidence for binding to the TACGGAAAT motif.

5. Page 11, line 225-227: 'We next searched for the 9-bp element within the RFT1 promoter, which is another direct target of OsSUF4, and discovered a 7-bp conserved element (5'-CGGAAAT-3'). Please provide information about the exact position of this element in the RFT1 locus in relation to the analyzed fragments shown in Fig. 4a.

6. Firefly-luciferase-based reporter assay (Fig. 6a). Normalization is done against 'vector'. I assume that this is a transfection with an empty vector bone?? The right control for normalization would be the same individual reporter constructs, but then without co-transfection with the effector construct.

7. Page 30, line 264: 'Of these, 100 genes (75.8%) showed enriched H3K36me3 modification'. Is this a true enrichment? What is the percentage when random set of ~130 genes are checked for H3K36me3 enrichment out of the ~1700 downregulated genes not containing the 7-bp element? Please make this comparison and provide the numbers.

8. Page 13, line 265-267: 'Moreover, H3K36me3 enrichment of the 100 genes was significantly increased close to the TSS compared with other H3K36me3-enriched genes (n=16,046)...'. (1) How was significance calculated? (2) Please make clear to the reader that this is downstream of the TSS and not upstream, where the consensus binding site for OsSUF4 is located. Note that once more is shown that H3K36me3 deposition is not enriched in the promoter region for these specific examples, as was identified by the ChIP experiment shown in fig. 4!?!? (see comment 1 above).

9. Page 13: results of RNAseq analysis in the *suf4*Ri-1 mutant background in comparison to wild type. As far I can see, the list in the supplements doesn't include OsRFT1 and OsHd3a (or they are not annotated as such). How do the authors explain this?

10. In the first paragraphs of the discussion, new experiments are introduced and various conclusions about differences between (plant) species are based on these results. These results should be presented in the results session and exact set-up and design has to be described. For the complementation studies in Arabidopsis, the authors have to provide e.g. details on the number of lines analyzed and they should confirm at least that the identified transgenic lines have

ectopic expression of the transgene by qRT-PCR or other suitable method. The arabidopsis SUF4 promoter is e.g. used. Data has to be provided that this promoter gives suitable expression levels and patterns.

11. Page 17/18, line 363-364: 'SDG725, by interaction with the transcription factor OsSUF4, is enriched at promoter regions containing the OsSUF4-binding DNA element.'. On which basis do the authors conclude this? Please specify. I believe that only a small proportion of the genes targeted by SDG725 do have this specific SUF4-binding element.

12. Page 18: '.....indicating that SDG725 stabilizes OsSUF4 binding and subsequently activates the transcription of RFT1 and Hd3a'. How do the authors envision this at the molecular level? Is it stabilization of binding or can it also be another molecular mechanism? E.g. co-binding or a change in stoichiometry of SUF4 due to complex formation with the SDG protein?? The authors should discuss the various options in more detail.

Minor points:

- Page 2, line 29: 'gene' should be 'genes'.

- Page 3, line 48-52: 'In animal cells, H3K36me3 is mainly distributed at 3' ends of the gene body, but it is close to the transcription start site (TSS) in plants. This suggests that different mechanisms exist for the establishment of histone lysine methylations between plants and animals.' This definitely hints towards different functions of this mark in plants and animals, but not necessarily means that the deposition-mechanism is different.

- In supplemental figure 2, relative expression of OsSUF4 is plotted. Only, normalization is explained and not relative to which tissue expression is plotted. Please, explain.

- Page 9, line 173: 'Figure S2B' should be 'Figure S3'.

- Page 12: the authors refer to a 7-bp and a 9-bp element. Are these not the same? Please check and correct.

- Page 18, line 384: 'Alignment of OsSUF4 homologs with amino acid sequence was performed..'. I guess that the authors mean that alignments were made for amino acid sequences and not DNA. Sentence needs to be rephrased.

- Page 22, line 460: 'on media lacking tryptophan, threonine, and adenine (SD -T/-L/-A)'. Minus 'L' is not minus threonine. Please correct.

- Page 33, line 631: 'schematic of' should be 'schematic representation of'. Same holds for the legend of other figure panels in which a schematic representation of a locus is shown.

Reviewers' comments:

Reviewer #1 (Remarks to the Author):

The paper by Liu et al. report on the studies on targeting of H3K36me3 methyltransferase SDG725 by OsSUF4. After a long time studies of the epigenetic modification enzymes, the targeting mechanism becomes a hot topic recently. The global distribution of these enzymes to certain chromatin region precisely regulates the function of these enzymes. In this paper, the authors analyzed the global distribution of H3K36me3 from different species and found that plants possess a specific H3K36me3 pattern other than animals. Further, they identified the transcription factor OsSUF4 can target SDG725 to a 7-bp DNA element, shaping the global pattern of plant H3K36me3. The structural studies of the ZF of OsSUF4 potentially identified the DNA binding site. Generally, the paper is well ordered and of good quality. Some issues below can be considered to strengthen the paper.

Response: We appreciate the reviewer's encouragement on this work. Here below, we outline our responses to the reviewer's concerns in detail.

1. As the author only provided limited information, I do not quite understand the Fig 2b. I assume the author want to show the purification of GST-725C in the left panel which is stained by GST antibody and the pull-down of HA-OsSUF4 which is stained by HA antibody. The authors need to provide more details in the figure caption. Because the molecular sizes of GST-725C and HA-OsSUF4 are quite similar as indicated by the gel. The author need to provide the validation for the antibodies that HA antibody does not react with GST-725C and GST antibody does not react with HA-OsSUF4 by simply staining the raw proteins with the antibodies.

Response: We apologize for the mistake. Yes, the left panel in Fig. 2b showed purified GST-SDG725C (GST-725C) using the GST antibody, and the right panel showed HA-OsSUF4 pulled down by GST-SDG725C with the HA antibody. In the revised manuscript, we have corrected Fig. 2b and described clearly in the figure legend (Lines 654-658). According to the reviewer's suggestion, we validated the

antibodies against GST and HA in new Supplementary Figure 2. We proved that the antibody against HA does not react with GST-725C, and GST antibody does not react with HA-OsSUF4.

2. Fig 2c caption, more details. What is DIC, YN, and YC? Please provide the full name.

Response: Following the reviewer's suggestion, we have now included the full names of DIC, YN, and YC in the revised figure legend of Fig. 2c (Lines 660-661).

3. Fig 2d. On the top, it should be 'C2H2 type zinc finger domains'.

Response: We have corrected in the revised Fig. 2d.

4. Line163, briefly discuss about why Ghd7 and Hd1 were up regulated in suf4Ri-1 and 725Ri-1.

Response: Following the reviewer's suggestion, we have now included an explanation for the up-regulation of *Ghd7* in the revised manuscript (Lines 161-165). The transcript levels of *Hd1* were not obviously changed in *suf4Ri-1* and *725Ri-1* (p value > 0.05). The transcript levels of *Ghd7* did increase in *suf4Ri-1* and *725Ri-1*, in agreement with the previous studies showing that *Ghd7* functions as a repressor of flowering.

5. Fig 7a. Among the 1863 gene down regulated by suf4Ri-1, only 132 contain the 7-bp element. The ratio is quite low.

Response: The target genes of OsSUF4 will affect the transcription levels of their downstream genes, so there could be more genes down-regulated but without the 7-bp element in *suf4Ri-1*. Now we clarified that the genes without 7-bp might be indirectly down-regulated in the revised manuscript (Lines 291-294).

Does it contain the discussed key flowering genes, RFT1 and Hd3a? Please discuss.

Response: We used 14-day-old young rice seedlings in RNA-seq analysis. At this

stage, the transcription levels of flowering genes *RFT1* and *Hd3a* are almost undetectable. This explains the absence of *RFT1* and *Hd3a* from the 1863 down-regulated genes identified in *suf4Ri-1*. We have now clarified this in our revised manuscript (Lines 289-291).

6. Line 312, the binding affinity is different from the one in fig s7d.

Response: We apologize for the mistake. The *K_d* value should be $51.28 \pm 3.68 \mu\text{M}$, and we have corrected this error in the manuscript text (Line 342).

7. Line 560-6-562, supplement the reference for HKL2000, Phenix and Refmac. Which program for model building, Coot or some others? Please add it. Line561, Phenix but not Phoenix.

Response: In the revised manuscript, we have added the references for HKL2000, Phenix and Refmac, clarified that Coot were used for model building, and corrected the mistake for “Phenix” (Lines 605-608).

Reviewer #2 (Remarks to the Author):

In this manuscript a physical interaction was found between the rice SUF4 protein and SDG725 and various and extensive molecular and genetic experiments have been performed to decipher the putative molecular function of this protein complex. Based on the obtained results, the authors postulate that SUF4 targets SDG725 to promoter regions of a specific sub-set of genes in the rice genome, including the key flowering time regulators RFT1 and Hd3a. As a consequence, the H3K36me3 mark is supposed to be deposited at these loci resulting in enhanced expression of these selected genes. An interaction between SUF4-like proteins and SDG proteins, and targeting of SDG proteins to specific loci by this mechanism, has been shown previously in other species. Hence, this finding is not completely new. Furthermore, there are some inconsistencies and points of attention in the presented data, as summarized below:

Response: We thank the reviewer for helpful comments. As cited in our manuscript (Refs. 27 and 28), we have only found Arabidopsis SUF4 previously characterized from literatures. Our work on rice SUF4 brings substantial breakthroughs at both mechanistic and genome-wide levels. Here below, we outline our responses to the reviewer's concerns in detail.

1. The title: 'The transcription factor OsSUF4 interacts with SDG725 and promotes H3K36me3 establishment towards gene promoter regions' is not clearly reflecting the findings of the paper. H3K36me3 deposition is found in general towards the 5' end of the gene body and not in promoter regions. See also my more detailed comments on this aspect below.

Response: Following the reviewer's comment, we have modified the title to "The transcription factor OsSUF4 interacts with SDG725 in promoting H3K36me3 establishment".

2. For the various ChIP-PCR enrichment analyses three adjacent promoter regions have been selected in RFT1 and Hd3a loci (Fig. 4A).

(1) Please indicate the reasoning for selecting these specific region in the proximal promoter region? Theoretically, SUF4 could also bind somewhere else in the locus, or not?

Response: We have revised Fig. 4 by including more regions, covering both promoter and gene body. OsSUF4 was found to enrich at the promoter regions, but not within the gene body regions of *RFT1/Hd3a*.

(2) The exact resolution of this analysis is not clear from the schematic representation. Please indicate in figure 4 the distance between the three selected promoter regions and the distance in relation to the TSS.

Response: In the revised manuscript, we have added the scale bar on the diagram and we have also detailed the location in the body text (Lines 180-184).

(3) I miss local controls to exclude the possibility that the observed enrichments for the three promoter fragments are not due to specific binding by the Ab-targeted proteins, but e.g. due to a general increased accessibility of these promoter regions because of altered expression levels of these two loci in the mutant backgrounds. Please include a fragment in e.g. an intron or exon (towards the 3' end of the gene) as additional controls.

Response: Regions towards the 3'-end of *RFT1/Hd3a* have now been added. This indeed provides good negative controls. We thank the reviewer for this nice suggestion.

(4) H3K36me3 enrichment analysis is also done for this promoter region upstream of the TSS. According to the analysis presented in Fig. 1, genome-wide, this region is not enriched or even underrepresented in H3K36me3 deposition. Why was this specific region selected for the H3K36-trimethylation analysis in these two loci?

Response: We agree with the reviewer that our previous selection of only upstream regions of TSS does not provide a complete view of H3K36me3 distribution pattern. This has now been corrected by new data added for regions along gene body (revised

Fig. 4a). Our data indeed show a peak downstream of TSS, which is consistent with Fig. 1. Meanwhile, H3K36me3 was also detected at regions upstream of TSS albeit to lower levels.

(5) Note that for H3K36me3 enrichment, a comparison has to be made to an IP using a general H3 antibody, to ensure that observed differences are not just reflecting general differences in 'Histone3/chromatin' occupancy of the investigated genomic regions in particular genetic backgrounds.

Response: Following the reviewer's suggestion, we have performed ChIP experiments using the antibody against H3. The data have now been included as new Supplementary Fig. 7 in the revised manuscript. Similar H3 profiles on *RFT1/Hd3a* chromatin were observed in the wild type and the mutants, providing strong arguments that the decreases of OsSUF4, SDG725 and H3K36me3 at *RFT1/Hd3a* we observed in the mutants are not due to the changes of nucleosome occupancy. This has now been clarified in our revised manuscript (Lines 193-197).

3. Page 10, line 198-202: 'To further investigate the genetic relationship between OsSUF4 and RFT1/Hd3a, mutants containing a 1-bp thymine insertion within the coding region of RFT1 (known as rft1) and a 1-bp adenine insertion within that of Hd3a (hd3a), which respectively resulted in an early termination, were generated by clustered regularly interspaced short palindromic repeats (CRISPR)/CRISPR-associated protein (Cas)9.'. This reads as that an approach was followed to specifically create these two single nucleotide insertions. If so, please describe how this was done? Or did the authors perform a classical CRISPR-based mutagenesis and selected these two specific lines for further analyses? Please describe in more detail how these two mutant lines were generated and identified.

Response: We apologize for the confusion. As the reviewer said, we performed a classical CRISPR-based mutagenesis and selected these two specific lines for further analyses. Now, we have clarified and included the detailed information in the revised manuscript (Lines 198-208).

4. EMSA experiments (Fig. 5). I find the results presented in figure 5b puzzling and not convincing because of a lot of background and overexposure, making it difficult to compare exact patterns in the various lanes on the gel. How do the authors explain e.g. that the smaller protein domain (OsSUF4N) results in a larger shift in combination with DNA probe 'c' than the full length OsSUF4 protein? (when I check the pattern for the free 'c' probe in the two gels, the running conditions seem to be more or less the same and hence, this cannot be the cause of the observed large difference in shift). In contrast, the data presented in 5c are clear and providing evidence for binding to the TACGGAAAT motif.

Response: We agree with the reviewer and improve the EMSA experiments. In the revised manuscript, we replaced the Fig.5b with a new one, in which the samples containing the same DNA probe were separated on one gel. As shown in the middle panel of the new Fig.5b, the full-length OsSUF4 clearly resulted in a larger shifted band than the smaller protein domain OsSUF4N did.

5. Page 11, line 225-227: 'We next searched for the 9-bp element within the *RFT1* promoter, which is another direct target of OsSUF4, and discovered a 7-bp conserved element (5'-CGGAAAT-3'). Please provide information about the exact position of this element in the *RFT1* locus in relation to the analyzed fragments shown in Fig. 4a.

Response: Now we indicated the exact position of the 7-bp element (Nucleotides -989 to -983) within *RFT1* promoter in the revised manuscript (Lines 236-237) and highlighted the 7-bp elements at *RFT1/Hd3a* in the revised Fig.4a.

6. Firefly-luciferase-based reporter assay (Fig. 6a). Normalization is done against 'vector'. I assume that this is a transfection with an empty vector bone?? The right control for normalization would be the same individual reporter constructs, but then without co-transfection with the effector construct.

Response: As the reviewer mentioned, we did use the same individual reporter construct without co-transfection with the effector construct as the control for

normalization of each sample. Now we clarified the control for normalization in the revised legend of Fig. 6a.

7. Page 30, line 264: ‘Of these, 100 genes (75.8%) showed enriched H3K36me3 modification’. Is this a true enrichment? What is the percentage when random set of ~130 genes are checked for H3K36me3 enrichment out of the ~1700 downregulated genes not containing the 7-bp element? Please make this comparison and provide the numbers.

Response: Among 1731 down-regulated genes not containing the 7-bp element, 1000 sets of 132 randomly selected genes were checked for H3K36me3 enrichment analyses. The numbers of H3K36me3 enriched genes from 1000 sets are shown in histogram as below. The percentage for the gene number over 100 is 2.6% (26 times among 1000 sets), so the P value < 0.05 .

8. Page 13, line 265-267: ‘Moreover, H3K36me3 enrichment of the 100 genes was significantly increased close to the TSS compared with other H3K36me3-enriched genes ($n=16,046$)...’.

(1) How was significance calculated?

Response: The P value was calculated by the Kolmogorov–Smirnov test and we have

clarified it in the revised legend of Fig. 7.

(2) Please make clear to the reader that this is downstream of the TSS and not upstream, where the consensus binding site for OsSUF4 is located. Note that once more is shown that H3K36me3 deposition is not enriched in the promoter region for these specific examples, as was identified by the ChIP experiment shown in fig. 4!?!? (see comment 1 above).

Response: We agree with the reviewer, in the revised manuscript we have clarified as downstream of TSS. (Line 296).

9. Page 13: results of RNAseq analysis in the suf4Ri-1 mutant background in comparison to wild type. As far I can see, the list in the supplements doesn't include OsRFT1 and OsHd3a (or they are not annotated as such). How do the authors explain this?

Response: To search more potential target genes of OsSUF4, we used 14-day-old young seedlings for our RNA-seq analysis. At this early age of plants, the flowering genes *RFT1* and *Hd3a* are almost not expressed, this explains their absence from the list of down-regulated genes identified by RNA-seq in the *suf4Ri-1* mutant.

10. In the first paragraphs of the discussion, new experiments are introduced and various conclusions about differences between (plant) species are based on these results. These results should be presented in the results session and exact set-up and design has to be described. For the complementation studies in Arabidopsis, the authors have to provide e.g. details on the number of lines analyzed and they should confirm at least that the identified transgenic lines have ectopic expression of the transgene by qRT-PCR or other suitable method. The Arabidopsis SUF4 promoter is e.g. used. Data has to be provided that this promoter gives suitable expression levels and patterns.

Response: Following the reviewer's suggestion, in the revised manuscript we have moved those results in the Results part (Lines 263-279), and the detailed information

such as the SALK line number and the promoter length have been added in Methods part (Lines 485-491). We obtained 15 and 10 transgenic lines expressing *AtSUF4* or *OsSUF4* driven by the native promoter and terminator of *AtSUF4* ($P_{AtSUF4}::AtSUF4$ or $P_{AtSUF4}::OsSUF4$) in *suf4-2* background, respectively. By qRT-PCR, we confirmed that the expression levels of *OsSUF4* in two independent transgenic lines of $P_{AtSUF4}::OsSUF4$ were roughly similar to that of *AtSUF4* in a line of $P_{AtSUF4}::AtSUF4$, shown now in the new Supplementary Fig. 8b. These data clearly indicate that both *AtSUF4* and *OsSUF4* are equally expressed under the P_{AtSUF4} promoter. Because *FLC* expression in *suf4-2* was rescued by $P_{AtSUF4}::AtSUF4$ but not by $P_{AtSUF4}::OsSUF4$ in the revised Supplementary Fig. 8c, we concluded that *OsSUF4* fails to work in *Arabidopsis*.

11. Page 17/18, line 363-364: 'SDG725, by interaction with the transcription factor OsSUF4, is enriched at promoter regions containing the OsSUF4-binding DNA element.'. On which basis do the authors conclude this? Please specify. I believe that only a small proportion of the genes targeted by SDG725 do have this specific SUF4-binding element.

Response: We have taken the caution and revised the text as “SDG725, by interaction with the transcription factor OsSUF4, is enriched close to the promoter regions of some OsSUF4-targeted genes, such as *RFT1* and *Hd3a*.” (Lines 395-397).

12. Page 18: '.....indicating that SDG725 stabilizes OsSUF4 binding and subsequently activates the transcription of RFT1 and Hd3a'. How do the authors envision this at the molecular level? Is it stabilization of binding or can it also be another molecular mechanism? E.g. co-binding or a change in stoichiometry of SUF4 due to complex formation with the SDG protein?? The authors should discuss the various options in more detail.

Response: We have now revised our discussion by including various possible options (Lines 399-408).

Minor points:

- Page 2, line 29: 'gene' should be 'genes'.

Response: Now we have corrected the error in the revised manuscript.

- Page 3, line 48-52: *'In animal cells, H3K36me3 is mainly distributed at 3' ends of the gene body, but it is close to the transcription start site (TSS) in plants. This suggests that different mechanisms exist for the establishment of histone lysine methylations between plants and animals.'* This definitely hints towards different functions of this mark in plants and animals, but not necessarily means that the deposition-mechanism is different.

Response: We changed the sentence as "It suggests that different mechanisms for H3K36me3 establishment and function may exist between plants and animals" in the revised manuscript (Lines 43-44).

- In supplemental figure 2, relative expression of *OsSUF4* is plotted. Only, normalization is explained and not relative to which tissue expression is plotted. Please, explain.

Response: For analyzing the transcription levels of *OsSUF4* in different tissues, we used *OsUbiquitin* as an internal control gene, and the relative transcript shown in the Fig. S2 was the value of *OsSUF4* in different tissues dividing by the respective value of *OsUbiquitin*. We have clarified it in the revised supporting data.

- Page 9, line 173: 'Figure S2B' should be 'Figure S3'.

Response: The error has been corrected.

- Page 12: the authors refer to a 7-bp and a 9-bp element. Are these not the same? Please check and correct.

Response: The 9-bp (5'-TACGGAAAT-3') element was first found in *Hd3a* promoter by EMSA analysis, and then be further narrowed down to 7-bp (5'-CGGAAAT-3') according to the element within *RFT1* promoter. The 9-bp element contains the 7-bp

element.

- Page 18, line 384: *'Alignment of OsSUF4 homologs with amino acid sequence was performed.'* I guess that the authors mean that alignments were made for amino acid sequences and not DNA. Sentence needs to be rephrased.

Response: Following the reviewer's suggestion, we have rephrased the sentence in the revised manuscript (Lines 425-426).

- Page 22, line 460: *'on media lacking tryptophan, threonine, and adenine (SD -T/-L/-A).'* Minus 'L' is not minus threonine. Please correct.

Response: We apologize for this mistake. We have corrected the error in the revised manuscript (Line 505).

- Page 33, line 631: *'schematic of' should be 'schematic representation of'. Same holds for the legend of other figure panels in which a schematic representation of a locus is shown.*

Response: We have corrected it in the related legends in the revised manuscript and in the supporting data. Again, we thank the reviewer for helpful and detailed comments on our manuscript.

REVIEWERS' COMMENTS:

Reviewer #1 (Remarks to the Author):

All my concerns have been satisfied. No additional comments.

Reviewer #2 (Remarks to the Author):

In general, I am satisfied with the detailed response by the authors to the reviewer comments and the modifications and improvements made to the manuscript.

Point 7: Please add the analysis performed in response to this comment as supplementary data to the final manuscript.

Furthermore, one major issue remains: Why is the deposition of the H3K36me3 mark at the RFD1 and Hd3a locus so different from other OsSUF4-SDG725 targets? Compare data presented in Figure 4a (effect on H3K36me3 deposition both up and downstream of the TSS) with the data presented in Figure 7c (effect only downstream of TSS and when significant upstream, it points even in the other direction!?). This observation is at least puzzling. It will probably need a complete new study to explain this obvious contradicting response between different OsSUF4 targets and this might be out of scope for this manuscript, but the authors should at least describe and discuss this remarkable observation/result.

Minor points:

- Line 19: 'showed' should be 'show'.
- Line 24, 25: 'Our results reveal the regulatory mechanism by which H3K36me3 establishes close to gene promoter regions in plants.' Please be more precise. As shown in Fig. 1, genome wide H3K36me3 deposition is specifically and strongly enriched in the 5'-end of the gene body in rice. Hence, downstream of the TSS in the gene body! Alternatively, the authors can refer to the seemingly different situation for the RFD1 and Hd3a loci, in which indeed H3K36me3 deposition seems to be affected both up- and downstream of the TSS.
- Line 263, 264: 'As the counterparts of SDG725 and OsSUF4 in Arabidopsis, the interaction between AtSUF4 and SDG8 was previously reported.' Please mention the genes in the same order (first SDG8 and then AtSUF4).
- Line 389-391: 'Thus, SDG725/SDG8 could be recruited to OsSUF4/AtSUF4-associated promoters for the deposition of H3K36 methylation.' In my opinion, this cannot be concluded. As shown by the authors, recruitment of SDG725 and OsSUF4 to targets seems to be mutually dependent. Hence it cannot be concluded that one protein recruits the other to specific targets. A plausible explanation for the observations could be that upon interaction between the two proteins a complex is formed that can bind/target specific target sequences (as discussed by the authors further on in the discussion).
- Line 392-395: 'Because most genes encoding histone-modifying enzymes are widely expressed in plants, the enzyme transcription factor regulatory mechanism seems to be critical not only for enzyme targeting but also for their downstream functions'. Generally, TFs can provide specificity to broadly expressed histone-modifying enzyme encoding genes. However, this seems not to be the case here, whereas also OsSUF4 appears to be broadly expressed! Please, modify.
- Line 505. Please not that the one-letter code for tryptophan is 'W' and not 'T'.

REVIEWERS' COMMENTS:

Reviewer #1 (Remarks to the Author):

All my concerns have been satisfied. No additional comments.

Reviewer #2 (Remarks to the Author):

In general, I am satisfied with the detailed response by the authors to the reviewer comments and the modifications and improvements made to the manuscript.

Point 7: Please add the analysis performed in response to this comment as supplementary data to the final manuscript.

Response: Following the reviewer's suggestion, we have added this analysis as Supplementary Fig. 9 in the revised manuscript (Lines 298-304).

Furthermore, one major issue remains: Why is the deposition of the H3K36me3 mark at the *RFD1* and *Hd3a* locus so different from other *OsSUF4*-*SDG725* targets? Compare data presented in Figure 4a (effect on H3K36me3 deposition both up and downstream of the TSS) with the data presented in Figure 7c (effect only downstream of TSS and when significant upstream, it points even in the other direction!?). This observation is at least puzzling. It will probably need a complete new study to explain this obvious contradicting response between different *OsSUF4* targets and this might be out of scope for this manuscript, but the authors should at least describe and discuss this remarkable observation/result.

Response: Following the reviewer's suggestion, in the revised manuscript, we have described the observation that H3K36me3 enrichment on *RFT1* and *Hd3a* was also observed upstream of the TSS (fragment 4 for *RFT1* and fragment 12 for *Hd3a*), although the most enrichment of H3K36me3 are located downstream of the TSS (fragment 5 for *RFT1* and fragment 13 for *Hd3a*) (Lines 191-196).

Minor points:

- Line 19: ‘showed’ should be ‘show’ .

Response: We have corrected in the revised manuscript.

- Line 24, 25: ‘Our results reveal the regulatory mechanism by which H3K36me3 establishes close to gene promoter regions in plants.’ Please be more precise. As shown in Fig. 1, genome wide H3K36me3 deposition is specifically and strongly enriched in the 5’ -end of the gene body in rice. Hence, downstream of the TSS in the gene body! Alternatively, the authors can refer to the seemingly different situation for the RFD1 and Hd3a loci, in which indeed H3K36me3 deposition seems to be affected both up- and downstream of the TSS.

Response: We have modified the sentence precisely as “Our results reveal the regulatory mechanism by which H3K36me3 establishes at the 5' end of gene body in plants” (Lines 24-25).

- Line 263, 264: ‘As the counterparts of SDG725 and OsSUF4 in Arabidopsis, the interaction between AtSUF4 and SDG8 was previously reported.’ Please mention the genes in the same order (first SDG8 and then AtSUF4).

Response: We have corrected in the revised manuscript.

- Line 389-391: ‘ Thus, SDG725/SDG8 could be recruited to OsSUF4/AtSUF4-associated promoters for the deposition of H3K36 methylation.’ In my opinion, this cannot be concluded. As shown by the authors, recruitment of SDG725 and OsSUF4 to targets seems to be mutually dependent. Hence it cannot be concluded that one protein recruits the other to specific targets. A plausible explanation for the observations could be that upon interaction between the two proteins a complex is formed that can bind/target specific target sequences (as discussed by the authors further on in the discussion).

Response: Following the reviewer’s suggestion, we have changed the sentence as “Thus, OsSUF4/AtSUF4 help SDG725/SDG8 to target the 5’ end of gene body regions and promote H3K36 methylation establishment, which is in line with the

apparent TSS-proximal pattern of H3K36me3 in plants” (Lines 389-391).

Line 392-395: ‘Because most genes encoding histone-modifying enzymes are widely expressed in plants, the enzyme transcription factor regulatory mechanism seems to be critical not only for enzyme targeting but also for their downstream functions’ . Generally, TFs can provide specificity to broadly expressed histone-modifying enzyme encoding genes. However, this seems not to be the case here, whereas also OsSUF4 appears to be broadly expressed! Please, modify.

Response: We have corrected the sentence as “The interaction between enzymes and transcription factors might influence the targeting of the enzymes and the distribution of the corresponding modifications” (Lines 391-393).

- Line 505. Please note that the one-letter code for tryptophan is ‘W’ and not ‘T’.

Response: We apologize for the mistake. We have corrected this error in the revised manuscript (Line 508).